# Oral administration of *Blautia wexlerae* ameliorates obesity and type 2 diabetes via metabolic remodeling of the gut microbiota

Koji Hosomi[1,2], Mayu Saito[1,3], Jonguk Park [4], Haruka Murakami[5,6], Naoko Shibata[7], Masahiro Ando[7], Takahiro Nagatake[1], Kana Konishi[5,8], Harumi Ohno[5,9], Kumpei Tanisawa[5,10], Attayeb Mohsen[4], Yi-An Chen [4], Hitoshi Kawashima [4], Yayoi Natsume-Kitatani [11], Yoshimasa Oka[1,3], Hidenori Shimizu[1,3], Mari Furuta[1,2], Yoko Tojima[1,2], Kento Sawane[1,12], Azusa Saika[1,2], Saki Kondo[1,2], Yasunori Yonejima[1,3], Haruko Takeyama[7,13,14,15], Akira Matsutani[16], Kenji Mizuguchi [4,17], Motohiko Miyachi[5,10] & Jun Kunisawa [1,2,7,12,18,19,20,21,22] ✉

The gut microbiome is an important determinant in various diseases. Here we perform a cross-sectional study of Japanese adults and identify the *Blautia* genus, especially *B. wexlerae*, as a commensal bacterium that is inversely correlated with obesity and type 2 diabetes mellitus. Oral administration of *B. wexlerae* to mice induce metabolic changes and anti-inflammatory effects that decrease both high-fat diet–induced obesity and diabetes. The beneficial effects of *B. wexlerae* are correlated with unique amino-acid metabolism to produce S-adenosylmethionine, acetylcholine, and L-ornithine and carbohydrate metabolism resulting in the accumulation of amylopectin and production of succinate, lactate, and acetate, with simultaneous modification of the gut bacterial composition. These findings reveal unique regulatory pathways of host and microbial metabolism that may provide novel strategies in preventive and therapeutic approaches for metabolic disorders.

Because of its rising prevalence and associated comorbidities such as type 2 diabetes mellitus (T2DM), obesity is a major public-health concern[1]. T2DM, which is caused by genetic background, overnutrition, and other environmental factors, is a global epidemic disease that is characterized by inflammation, metabolic disorders such as insulin resistance, and β-cell dysfunction[2]. The gut microbiota is recognized as a key environmental factor that contributes to the pathophysiology of obesity[3–5] and T2DM[6–8]. For example, the gut microbiota produces a wide range of metabolites, some of which—such as lipopolysaccharide and tryptophan-derived metabolites—appear to be causative disease factors[9–11]. In contrast to the gut microbiota's disease-promoting effects, the short-chain fatty acids (SCFAs) and secondary bile acids generated by microbes are known to have anti-obesity and anti-diabetes properties[12–15].

Consistent with the pivotal role of the gut microbiota in the control of adipose tissue, obese persons and patients with T2DM have an altered gut microbial composition[16]. For example, diabetic European women demonstrate alterations in the composition and function of the gut microbiota[7]. Of note, the mathematical model established from their metagenomic profiles identified T2DM with high accuracy among Europeans—but not Chinese people—indicating that the discriminant metagenomic markers for T2DM differ between European and Chinese populations[7]. Several lines of evidence have indicated regional differences in gut microbial composition[17,18], which are influenced by various environmental factors including diet[19–21] and genetic background[5,22]. However, the mechanisms underlying the effects of differences in gut microbial composition on obesity and T2DM remain poorly understood.

Japanese people have a unique dietary culture and habits, and their gut microbiome shows greater abundance in the genera *Bifidobacterium* and *Blautia* compared with those of people in other countries[18]. Several studies have investigated the gut microbiota of Japanese subjects and its association with obesity and T2DM. For example, one group[23] used 16S rRNA amplicon sequencing to analyze 10 subjects (4 non-obese and 6 obese), and others[24,25] applied a reverse transcription-quantitative PCR method to evaluate Japanese patients with T2DM (50 or 19 subjects). In addition to their unique gut microbial composition, Japanese people exhibit the highest average life span worldwide and a very low body mass index (BMI)[26]; therefore a study involving Japanese participants may increase our understanding of the gut microbiota and its contributions to ameliorating obesity and T2DM.

In this work, we conducted a cross-sectional study of Japanese adults and found that the abundance of *Blautia wexlerae* was inversely correlated with obesity and T2DM. In addition, the administration of *B. wexlerae* causatively decreased high-fat diet-induced obesity and diabetes in mice. Furthermore, we identified several unique metabolites of *B. wexlerae* that altered energy metabolism, exerted anti-inflammatory effects, and changed the composition of the gut bacterial environment.

## Results

### Japanese cohort study

We conducted a cross-sectional study involving 217 participants as a discovery cohort to assess the association between the gut microbiota and obesity or T2DM in Japanese adults (Supplementary Table 1, 2). Multiple-regression analysis revealed high correlation between the gut microbiota and BMI ($R^2 = 0.32$) and T2DM ($R^2 = 0.29$). Multiple-regression analysis to explore gut microbiome alterations in obesity identified 16 genera that were related to BMI (Supplementary Table 3), among which 11 organisms were selected according to the R-squared score from single regression analysis (Fig. 1A). In the same way, combining multiple logistic regression analysis and single logistic regression analysis found 8 organisms that were differentially abundant in samples from T2DM patients compared with control subjects (Fig. 1B, Supplementary Table 4). Obesity (BMI ≥ 25 in accordance with the guideline of the Japan Society for the Study of Obesity[27]) and T2DM were commonly associated with decreased abundance of *Blautia*, *Faecalibacterium*, and *Butyricicoccus* and an increase in *Megasphaera* abundance (Fig. 1C, Supplementary Fig. 1A, B). *Faecalibacterium* and *Butyricicoccus* are known butyrate-producing bacteria[16,28], and previous studies showed potential beneficial roles of *Faecalibacterium* in both human cohorts and animal models[16].

Although several studies have implicated the involvement of *Blautia* in the development of obesity and T2DM[16,29], little definitive information is available. We therefore assessed the effect of *Blautia* on obesity and T2DM via evaluating the odds ratio (OR) and learned that the *Blautia* abundance was inversely correlated to the ORs of obesity and T2DM (Fig. 1D, Supplementary Table 5 and 6). This finding was supported by quantitative PCR data, which showed a similar tendency regarding the relative abundance of *Blautia* according to 16S rRNA gene amplicon sequencing analysis (Supplementary Fig. 2A–C).

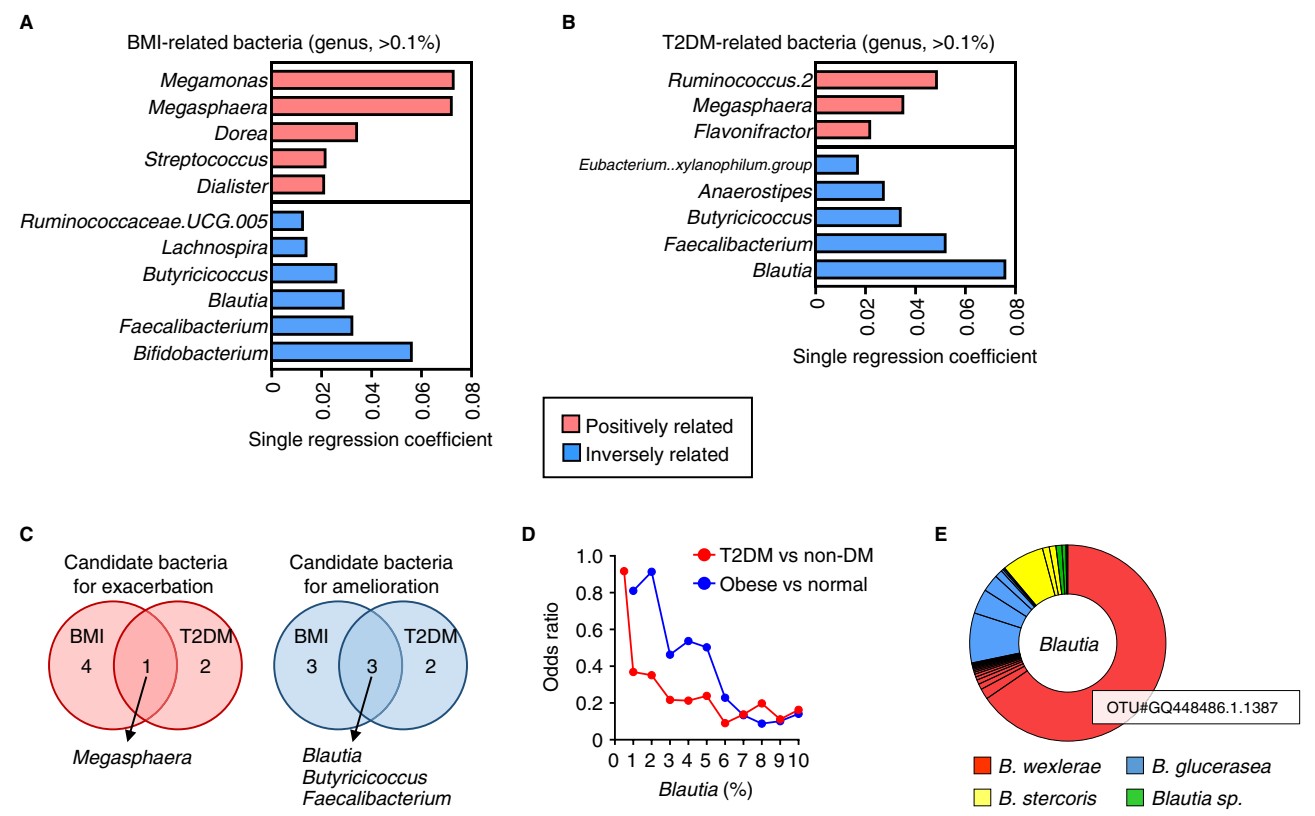

**Fig. 1 | Intestinal bacterial genera associated with body mass index (BMI) and type 2 diabetes mellitus (T2DM) in Japanese adults. A** BMI-related bacterial genera, which were selected and ranked according to $R^2$ score from single regression analysis ($P < 0.05$) among 16 genera that were identified through multiple-regression analysis by forward selection (Supplementary Table 3) by using the data of the 217 participants (Supplementary Table 1). **B** T2DM-related bacterial genera, which are selected and ranked according to $R^2$ score from single regression analysis ($P < 0.05$) among 22 genera that were identified through multiple logistic regression analysis by forward selection (Supplementary Table 4) by using the data of 192 participants (comprising 147 nonDM subjects and 45 T2DM patients and excluding 25 patients with Type 1 diabetes) (Supplementary Table 1). **C** Intestinal genera associated with both BMI and T2DM. **D** Odds ratios for *Blautia* abundance in the development of obesity (BMI ≥ 25) and T2DM. **E** Estimation of *Blautia* species according to BlastN analysis of representative OTU sequences.

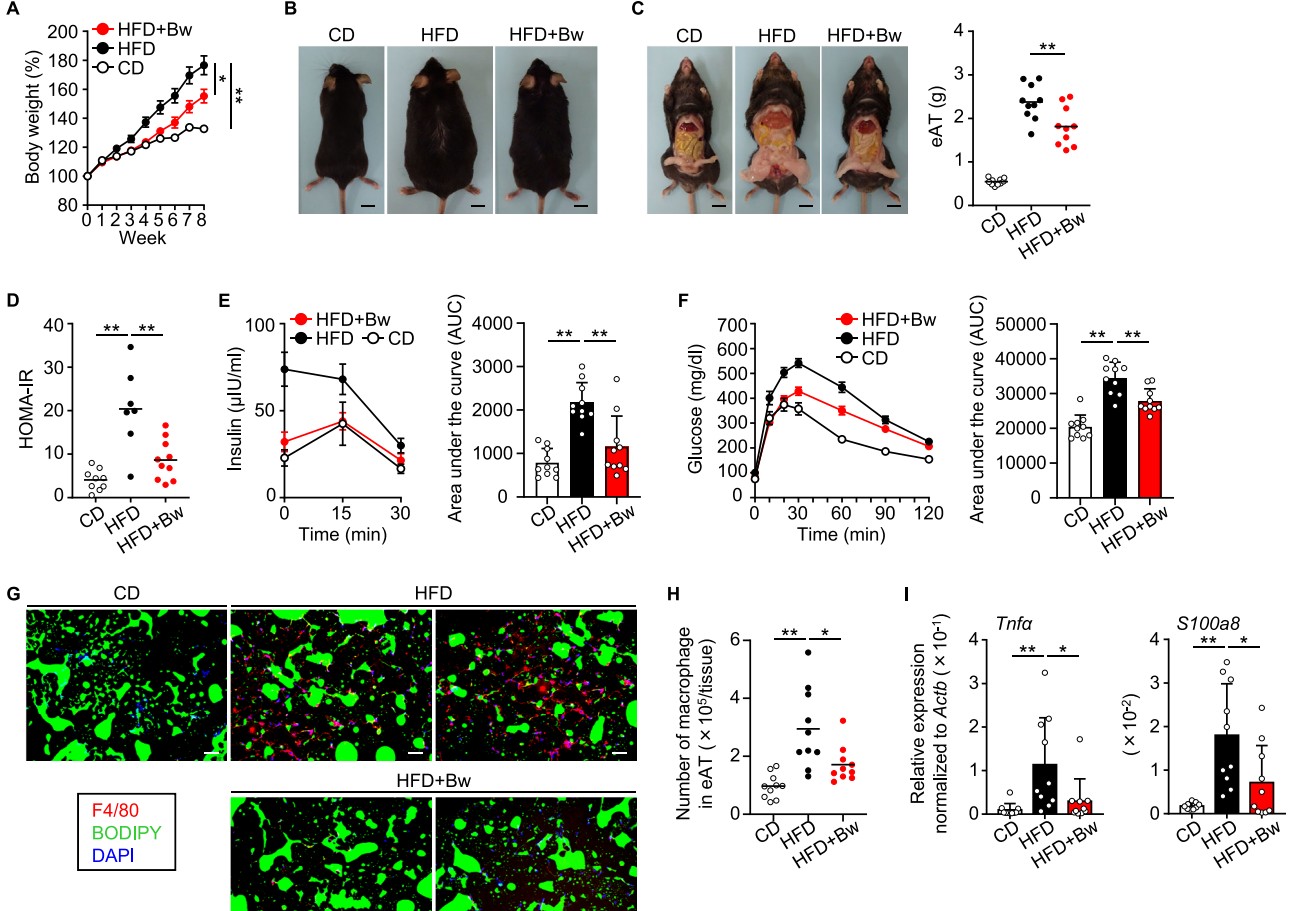

**Fig. 2 | High-fat diet (HFD)-induced obesity and diabetes were ameliorated by oral administration of *Blautia wexlerae* to mice. A** Mice were maintained on standard chow (control diet, CD) or HFD without or with oral administration of *B. wexlerae* (Bw) three times each week and were weighed weekly. Data are representative of at least three independent experiments ($n = 5$, mean ± 1 SD). *$P = 0.0105$; **$P = 0.0005$ (two-way ANOVA). **B** Photographs of representative mice. **C** Photographs of representative mice and weight of epididymal adipose tissue (eAT). Data are combined from two independent experiments ($n = 10$, mean). **$P = 0.0039$ (one-way ANOVA). **D** HOMA-IR, an indicator of insulin resistance, calculated as 'glucose (mg/dl) × insulin (μU/ml)/405'. Data are combined from two independent experiments without hemolytic samples ($n = 7$–10, mean). **$P < 0.01$ (one-way ANOVA). **E** Blood insulin was monitored through intraperitoneal glucose tolerance testing (IPGTT). Data are combined from two independent experiments

($n = 10$, mean ± 1 SD). **$P < 0.01$ (one-way ANOVA). **F** Blood glucose was monitored by using IPGTT. Data are combined from two independent experiments ($n = 10$, mean ± 1 SD). **$P < 0.01$ (one-way ANOVA). **G** Representative immunohistologic analysis of eAT. Macrophages, lipid droplets, and nuclei were visualized by using F4/80 monoclonal antibody (red), BODIPY (green), and DAPI (blue) staining, respectively. Scale bar, 100 μm. **H** Number of macrophages in the eAT. Data are combined from two independent experiments ($n = 10$, mean). *$P = 0.0140$; **$P = 0.0001$ (one-way ANOVA). **I** Gene expression of *Tnfα*, an inflammatory cytokine, and *S100a8*, a chemokine for recruiting macrophages, in the eAT mature adipocyte fraction (MAF). Data are combined from two independent experiments ($n = 10$, mean ± 1 SD). *$P < 0.05$; **$P < 0.01$ (one-way ANOVA). CD, lean mice fed a standard chow diet (CD-fed mice); HFD, obese mice fed a high-fat diet (HFD-fed mice); HFD + Bw, HFD-fed mice orally supplemented with *B. wexlerae*.

Notably, because *Blautia* was highly discriminant for obesity and T2DM in our Japanese population, we subsequently focused this study on the role of *Blautia* in the control of obesity and diabetes.

We next investigated whether factors other than the prevalent obesity and T2DM, such as age, sex, and medication (e.g., metformin), influenced the proportion of *Blautia*[30] but found no association between *Blautia* abundance and these factors (Supplementary Fig. 3). Furthermore, we tested the gut microbiome composition in nonDM and T2DM samples by using data sets randomly selected from study participants that were similarly distributed in terms of age and sex. *Blautia* abundance was significantly increased in the adjusted nonDM samples compared with T2DM samples (Supplementary Table 7), suggesting little confounding effect of factors such as age and sex on this analysis.

To confirm the association between *Blautia* abundance and obesity, we conducted another cross-sectional study involving 195

participants from a different region of Japan as a validation cohort (Supplementary Table 8). This validation study reproduced the results of our discovery study, specifically the decreased abundance of *Blautia* in obesity (Supplementary Fig. 4A) as well as little confounding effect of age and sex (Supplementary Fig. 4B, C).

We then used the sequences of representative operational taxonomic units (OTUs) as query strings in BLASTN searches to estimate which species of the *Blautia* genus were inversely associated with obesity and T2DM; this analysis returned three *Blautia* species—*B. wexlerae*, *B. glucerasea*, and *B. stercoris*. Among these, *B. wexlerae*, especially OTU GQ448486.1.1387, predominated, and the relative abundance of OTU GQ448486.1.1387 was nearly equal to that of *Blautia* overall (Fig. 1E, Supplementary Fig. 5). Together, these findings suggest *B. wexlerae* as a candidate gut microbe that might potently ameliorate obesity and T2DM.

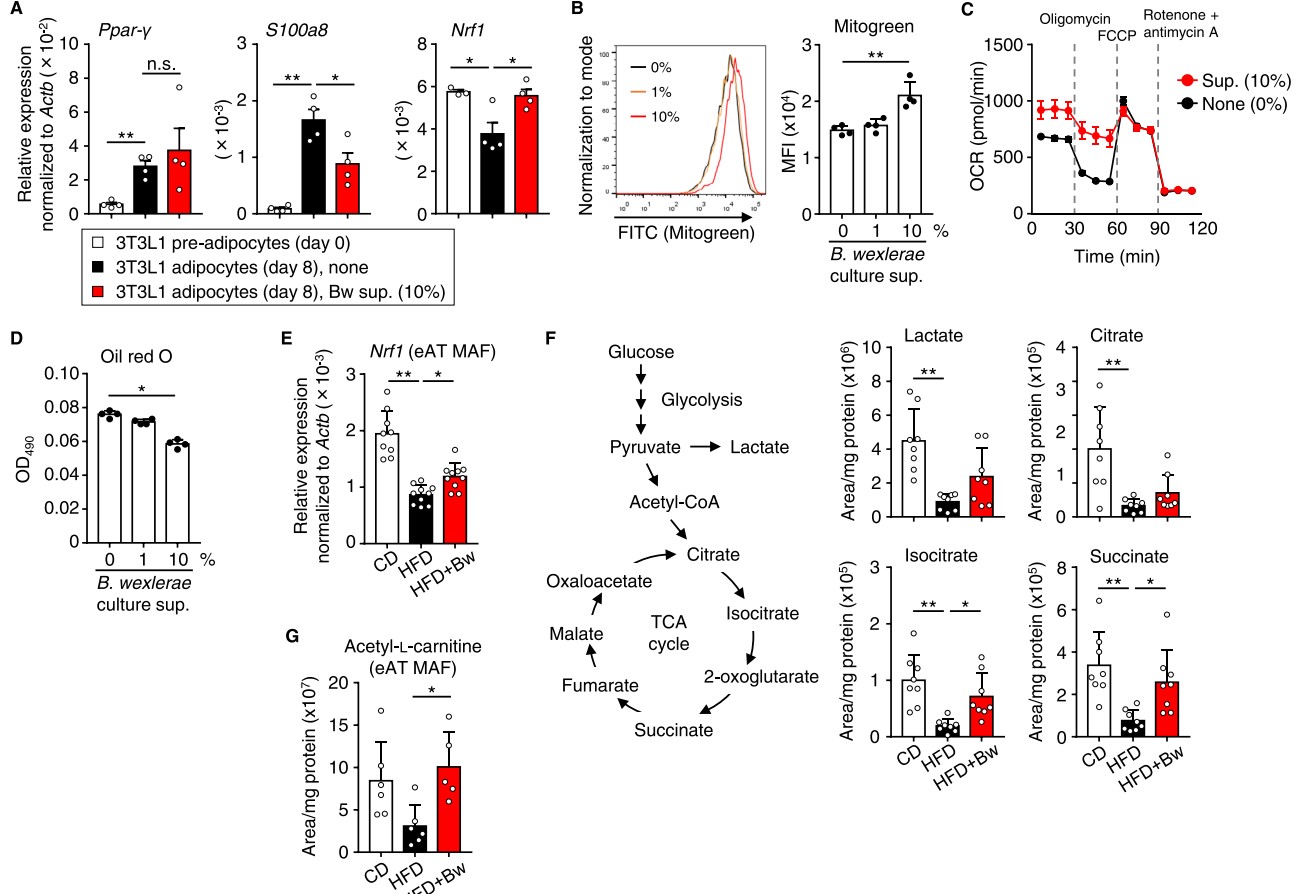

**Fig. 3 | *Blautia wexlerae*-derived metabolites showed anti-inflammatory and anti-adipogenesis properties in adipocytes as well as alteration of mitochondrial metabolism. A** 3T3L1 pre-adipocytes were differentiated into mature adipocytes in the absence (none) or presence of the cultured supernatant of *B. wexlerae* at a final concentration of 10%. Gene expression of *Pparγ* (a transcription factor used as a marker of adipocyte differentiation), *S100a8* (a chemokine for recruiting macrophages), and *Nrf1* (a transcriptional factor used as a marker of mitochondrial biogenesis) was measured in 3T3L1 pre-adipocytes and adipocytes. Data are representative of two independent experiments without samples below the detection limit ($n$ = 3–4, mean ± 1 SD). *$P$ < 0.05; **$P$ < 0.01; n.s. not significant (one-way ANOVA). **B** Mitochondrial mass was measured by flow cytometry analysis using Mitogreen in 3T3L1 adipocytes treated without or with culture supernatant (sup.) of *B. wexlerae* at a final concentration of 1% or 10%. Data are representative of two independent experiments ($n$ = 4, mean ± 1 SD). **$P$ = 0.0012 (one-way ANOVA). **C** By using an XF24 extracellular flux analyzer, the oxygen consumption rate (OCR) was measured in 3T3L1 adipocytes treated without or with the cultured supernatant of *B. wexlerae* at the concentration of 10%. Data are combined from two

independent experiments ($n$ = 14, mean ± 1 SD). **D** Lipid accumulation was assessed by using oil red O staining in 3T3L1 adipocytes treated without or with culture supernatant of *B. wexlerae* at a final concentration of 1% or 10%. Data are representative of two independent experiments ($n$ = 4, mean ± 1 SD). *$P$ = 0.0316 (one-way ANOVA). **E** Gene expression of *Nrf1* in the eAT MAF of mice. Data are combined from two independent experiments without samples below the detection limit ($n$ = 9–10, mean ± 1 SD). *$P$ = 0.0421; **$P$ < 0.0001 (one-way ANOVA). **F** Representative metabolites of glycolysis (lactate) and the TCA cycle (citrate, isocitrate, and succinate) in the eAT MAF of mice were measured by using liquid chromatography–tandem mass spectroscopy (LC–MS/MS). Data are combined from two independent experiments ($n$ = 8, mean ± 1 SD). *$P$ < 0.05; **$P$ < 0.01 (one-way ANOVA). **G** Acetyl-ʟ-carnitine, a constituent of the inner mitochondrial membrane, in the eAT MAF of mice was measured by using LC–MS/MS. Data are combined from two independent experiments without samples below the detection limit ($n$ = 5–6, mean ± 1 SD). *$P$ = 0.0231 (one-way ANOVA). CD CD-fed mice; HFD HFD-fed mice, HFD + Bw HFD-fed mice supplemented with *B. wexlerae*.

## Oral administration of *B. wexlerae* modulated the phenotypes of high-fat diet-induced obesity and diabetes in mice

Given the findings from our human cohort study, we then turned to using a murine model (Supplementary Fig. 6A) to examine the causality of *B. wexlerae* in obesity and diabetes. Compared with lean mice fed a standard chow diet (CD-fed mice), obese mice fed a high-fat diet (HFD-fed mice) had increased body weight, accumulation of peritesticular fat, and increased epididymal adipose tissue (eAT) weight (Fig. 2A–C). Oral supplementation with *B. wexlerae* had no effect on the mice's body weight gain prior to a HFD feeding term (Supplementary Fig. 6B) or on food or energy consumption (Supplementary Fig. 7A–C). However, oral administration of *B. wexlerae* during HFD feeding protected mice from becoming obese, concurrent with decreased body weight gain and fat accumulation, compared with nonsupplemented

HFD-fed mice (Fig. 2A–C). These results suggest that *B. wexlerae* has potential to contribute to the prevention of obesity.

We next examined the pathology associated with the obesity-induced diabetes. Compared with CD-fed mice, HFD-fed mice showed increased blood levels of glucose and insulin under fasting conditions and increased plasma HOMA-IR, an indicator of insulin resistance (Fig. 2D, Supplementary Fig. 8A, B); administration of *B. wexlerae* decreased these diabetes indicators in HFD-fed mice (Fig. 2D, Supplementary Fig. 8A, B). During intraperitoneal glucose tolerance testing (IPGTT), CD-fed mice showed the expected transient increases in blood insulin, followed by rapid return to the original level, after intraperitoneal injection of glucose (Fig. 2E). In comparison, glucose injection had no effect on blood insulin levels in HFD-fed mice (Fig. 2E), but HFD-fed mice supplemented with *B. wexlerae* showed similar

responses in blood insulin to those of CD-fed mice (Fig. 2E). Consistent with these results, the increases in blood glucose after intraperitoneal glucose injection in HFD-fed mice were normalized to the levels of CD-fed mice when HFD-fed mice were given *B. wexlerae* (Fig. 2F).

Inflammation in the eAT contributes to the development of obesity-induced diabetes. We therefore histologically analyzed the eAT in HFD-fed mice and found macrophages that were accumulated into 'crown-like structures', which are a histological hallmark of inflammation (Fig. 2G, Supplementary Fig. 9). Consistent with the inhibitory effects of *B. wexlerae* on diabetes pathogenesis, the eAT of HFD-fed mice that received *B. wexlerae* showed little macrophage accumulation (Fig. 2G, Supplementary Fig. 9). In addition, flow cytometric analysis showed that administration of *B. wexlerae* decreased the number of macrophages, especially pro-inflammatory M1-like macrophages, that infiltrated into the eAT of HFD-fed mice (Fig. 2H, Supplementary Fig. 10A, B). Along the same line, RT-qPCR analysis indicated that the expression of *Tnfα*, which encodes an inflammatory cytokine, and *S100a8*, which codes for a chemokine that recruits macrophages, was decreased in the mature adipocyte fraction (MAF) in the eAT of HFD-fed mice given *B. wexlerae* (Fig. 2I). Therefore, the administration of *B. wexlerae* inhibited not only body weight gain but also inflammatory responses in the eAT of HFD-fed mice, thus explaining—at least in part—the organism's inhibitory effects on diabetes.

## Metabolites from *B. wexlerae* inhibit inflammation and adipogenesis through their effects on adipocytes

The decreased expression of pro-inflammatory cytokines in adipocytes from mice prompted us to use in vitro differentiated adipocytes to examine whether *B. wexlerae* might alter their pro-inflammatory cytokine production. During the differentiation of 3T3L1 pre-adipocytes into adipocytes in vitro, we treated the cells with the supernatants from *B. wexlerae* cultures (Supplementary Fig. 11A). Upon differentiation, *B. wexlerae*-untreated 3T3L1 cells and those treated with *B. wexlerae*-cultured medium expressed similarly increased levels of *Pparγ*, a transcription factor used as a marker of adipocyte differentiation (Fig. 3A), suggesting that molecules produced by *B. wexlerae* and present in the culture supernatant did not affect adipocyte differentiation. Under these same experimental conditions, treatment of 3T3L1 adipocytes with *B. wexlerae*-cultured supernatant reduced the expression of *S100a8* with little effect on *Tnfα* expression (Fig. 3A, Supplementary Fig. 11B), thus indicating that *B. wexlerae*-derived metabolites inhibited S100a8 expression in adipocytes.

Several studies have suggested an association between impaired mitochondrial function and diabetes pathogenesis or inflammation[31,32]. We therefore assessed the gene expression of nuclear respiratory factor 1 (Nrf1), a transcriptional factor used as a marker of mitochondrial biogenesis[33]; the addition of *B. wexlerae*-cultured medium increased *Nrf1* expression in 3T3L1 adipocytes (Fig. 3A). Consistent with these gene expression data, flow cytometry using Mitogreen detected increased mitochondrial mass in 3T3L1 adipocytes treated with *B. wexlerae* culture supernatant (Fig. 3B). Functionally, treating 3T3L1 adipocytes with *B. wexlerae*-cultured medium increased mitochondrial basal respiratory oxygen consumption, characterized by increased proton leakage and decreased ATP synthesis, but did not affect glycolysis (Fig. 3C, Supplementary Fig. 12A, B), thus suggesting mitochondrial uncoupling for promoting energy expenditure. We also investigated lipid metabolism and found that, upon differentiation, 3T3L1 cells formed lipid drops in the cytoplasm, which were detected by oil red O staining. Consistent with the results of our in vivo experiments (Fig. 2C), the treatment of 3T3L1 adipocytes with *B. wexlerae*-cultured medium decreased lipid accumulation in a dose-dependent manner (Fig. 3D).

We then aimed to confirm these phenomena in vivo. Indeed, the expression of *Nrf1* was reduced in the eAT mature adipocyte fraction (MAF) of HFD-fed mice compared with CD-fed mice, and the

administration of *B. wexlerae* to HFD-fed mice increased *Nrf1* expression (Fig. 3E), suggesting altered mitochondrial metabolism in the obese mice. Therefore, we next examined energy metabolism through liquid chromatography–tandem mass spectrometry (LC–MS/MS) of eAT MAF. Signals associated with representative metabolites of glycolysis (lactate) and the TCA cycle (citrate, isocitrate, and succinate) were decreased in HFD-fed mice compared with CD-fed mice (Fig. 3F), indicating metabolic abnormalities associated with obesity and diabetes, similar to results from a previous study[34]. The administration of *B. wexlerae* to HFD-fed mice did not affect their lactate or citrate levels but increased their isocitrate and succinate levels (Fig. 3F), suggesting that *B. wexlerae* promotes energy metabolism in adipose tissue. This result was supported by the presence of increased acetyl-L-carnitine, a constituent of the inner mitochondrial membrane[35,36], in the eAT MAF of HFD-fed mice that received *B. wexlerae* (Fig. 3G). These findings collectively indicate that *B. wexlerae* produces metabolites that have potential to modify host inflammatory responses and energy metabolism.

Next, we explored this mechanism in other tissues, namely muscle and liver. Regardless of differences in diet or administration of *B. wexlerae*, glycolysis and TCA cycle-associated metabolites in gastrocnemius muscle were similar among all groups of mice (Supplementary Fig. 13A). In contrast, liver levels of lactate, citrate, and succinate were decreased in HFD-fed mice compared with CD-fed mice (Supplementary Fig. 13B), and the administration of *B. wexlerae* to HFD-fed mice did not alter lactate or citrate levels but increased their succinate level (Supplementary Fig. 13B). Thus, adipose tissue and liver demonstrated similar metabolic changes—in particular, succinate accumulation, which is a metabolic signature of thermogenesis[37]—suggesting that the administration of *B. wexlerae* promotes energy expenditure in the adipose tissue and liver of HFD-fed mice.

As other possible targets for the control of obesity and diabetes, the microbes modulate the host release of gut hormones such as GLP-1 to affect host metabolism and energy homeostasis[14], but the serum GLP-1 level did not differ between *B. wexlerae*-treated and -untreated HFD-fed mice (Supplementary Fig. 14). Another potential target is energy excretion. Reflecting the energy content of each diet, the energy excreted in feces was greater in HFD-fed mice than CD-fed mice (Supplementary Fig. 15A), but the administration of *B. wexlerae* did not alter energy excretion from HFD-fed mice (Supplementary Fig. 15A). In addition, HFD-fed mice showed similar levels of spontaneous activity regardless of the administration of *B. wexlerae* (Supplementary Fig. 15B).

## Metabolic pathways of *B. wexlerae* and their unique metabolites

To identify potent *B. wexlerae*-derived metabolites beneficial for controlling obesity and diabetes, we sought to reveal the metabolic uniqueness of *Blautia* on the basis of genetic information. Using iPATH3.0[38], we compared the presence of Kyoto Encyclopedia of Genes and Genomes (KEGG) orthologous groups[39] among *Blautia* (KEGG organism code: rob), *Bacteroides* (KEGG organism code: bvu), *Prevotella* (KEGG organism code: pru), and *Faecalibacterium* (KEGG organism code: fpr), which are major intestinal genera in our human data (Supplementary Fig. 16) and in healthy Japanese individuals[40] (Supplementary Figs. 17–19). This analysis revealed 61 KEGG orthologous groups that are unique to *Blautia*, and TargetMine identified 10 enriched pathways (*P* < 0.01, Benjamini–Hochberg correction) in the metabolic pathway map (Supplementary Fig. 20). Together, these results suggest various unique features of *Blautia* in regard to amino acid, carbohydrate, and nucleotide metabolism (Supplementary Fig. 21).

These findings prompted us to analyze the supernatant from *B. wexlerae* cultures by using LC–MS/MS. Data analysis of the resulting volcano plot revealed seven metabolites that were increased more than fourfold in the supernatant of *B. wexlerae* cultures compared with

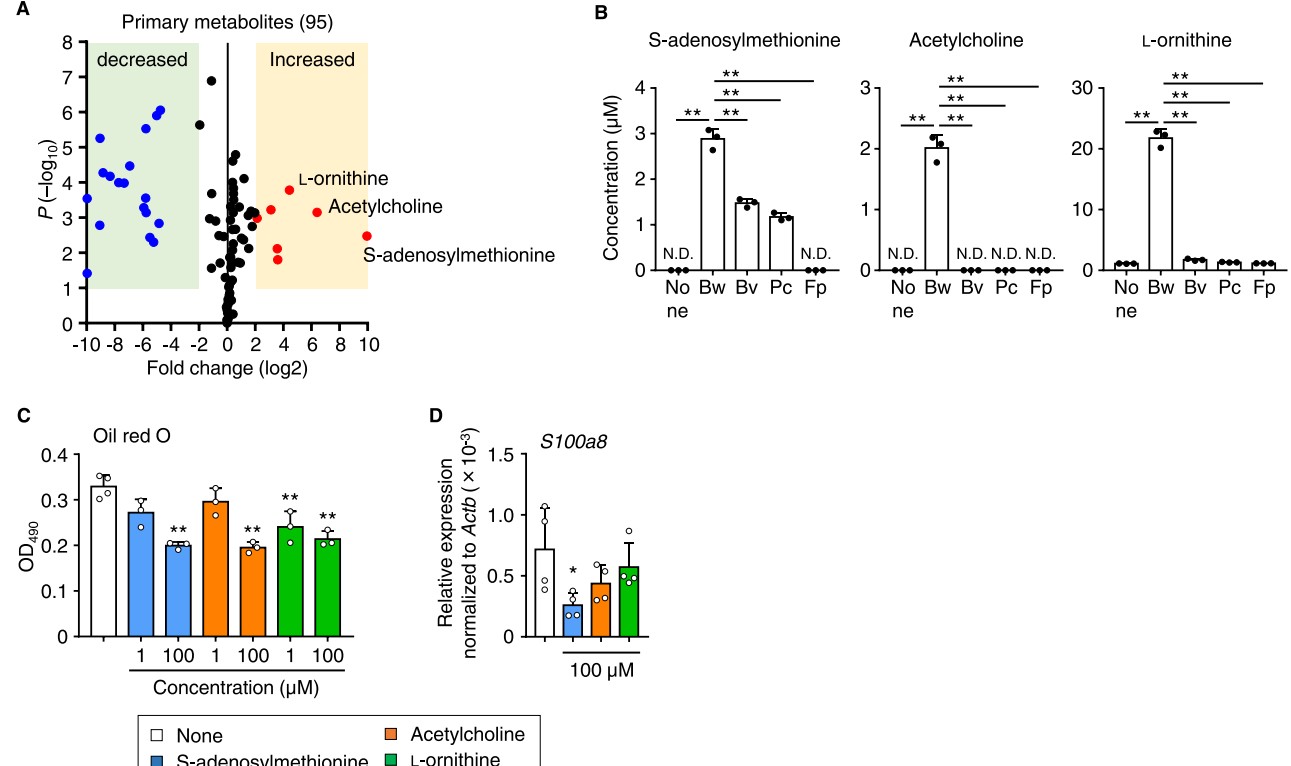

**Fig. 4 | *Blautia wexlerae* showed unique characteristics in amino acid meta-bolism, such as production of S-adenosylmethionine, acetylcholine, and ʟ-ornithine. A** Volcano plot showing LC–MS/MS analysis of *B. wexlerae* culture supernatant. Red and blue dots indicate metabolites increased and decreased, respectively, by more than fourfold in *B. wexlerae* culture supernatant compared with fresh medium (*n* = 4 biologically independent samples). Statistical significance was evaluated by using two-tailed unpaired *t*-test. **B** Quantitative measurement by LC–MS/MS of S-adenosylmethionine, acetylcholine, and ʟ-ornithine in fresh med-ium (none) and culture supernatants of *B. wexlerae* (Bw) and major intestinal bacteria including *Bacteroides vulgatus* (Bv), *Prevotella copri* (Pc), and

*Faecalibacterium prausnitzii* (Fp) (*n* = 3, mean ± 1 SD). ND not detected. **P < 0.01 (one-way ANOVA in comparison with none group). **C** Lipid accumulation was measured by oil red O staining in 3T3L1 adipocytes treated without (none) or with S-adenosylmethionine, acetylcholine, and ʟ-ornithine at the concentration of 1 or 100 μM (*n* = 3–4, mean ± 1 SD). **P < 0.01 (one-way ANOVA). **D** Gene expression of *S100a8*, a chemokine for recruiting macrophages, in 3T3L1 adipocytes treated without (none) or with S-adenosylmethionine, acetylcholine, and ʟ-ornithine at 100 μM (*n* = 4, mean ± 1 SD). *P = 0.0295 (one-way ANOVA in comparison with none group). Data are representative of two independent experiments (**A**–**D**).

uncultured, fresh medium (Fig. 4A). Furthermore, among these seven metabolites, high levels of S-adenosylmethionine, acetylcholine, and ʟ-ornithine were detected specifically in *B. wexlerae*-cultured super-natant compared with those from cultures of other major intestinal bacteria (i.e., *Bacteroides vulgatus*, *Prevotella copri*, *Faecalibacterium prausnitzii*) (Fig. 4B).

In the S-adenosylmethionine cycle, *B. wexlerae* appears to convert extracellular S-adenosylhomocysteine to S-adenosylmethionine via intermediates such as homocysteine and methionine (Supplementary Fig. 22A). Although other intestinal bacteria likewise uptake S-adenosylhomocysteine into the cell, its subsequent metabolism appears to differ among microbes (Supplementary Fig. 22A). The pathway in *F. prausnitzii* is unclear, but *B. vulgatus* and *P. copri* produce homocysteine and cysteine from S-adenosylhomocysteine rather than S-adenosylmethionine (Supplementary Fig. 22A).

Acetylcholine typically is biosynthesized from serine via ethano-lamine and choline[41]. Indeed, serine—but not choline—was decreased in *B. wexlerae*-cultured supernatant but not in that from other major intestinal bacteria, suggesting that *B. wexlerae* may uniquely utilize extracellular serine rather than choline to synthesize acetylcholine (Supplementary Fig. 22B). Furthermore, citrulline, arginine, and orni-thine are central metabolites of the urea cycle[42]. However, citrulline and arginine contents were decreased in the supernatant from cultures of *B. wexlerae* compared with other major intestinal bacteria, sug-gesting that *B. wexlerae* exclusively utilizes extracellular citrulline and arginine to produce ʟ-ornithine (Supplementary Fig. 22C). Therefore,

*B. wexlerae* has several unique metabolic characteristics regarding amino acid synthesis and consequently produces unique metabolites.

S-adenosylmethionine, acetylcholine, and ʟ-ornithine have anti-inflammatory properties and modify aspects of host metabolism, such as lipid metabolism, indicating that these compounds are potential effector metabolites for controlling obesity and diabetes. To address this hypothesis, we examined the effects of these metabolites on 3T3L1 adipocytes. Treatment of these cells with S-adenosylmethionine, acetylcholine, or ʟ-ornithine decreased lipid accumulation in a dose-dependent manner (Fig. 4C); S-adenosylmethionine also suppressed the expression of *S100a8* (Fig. 4D).

Despite several attempts, we were unable to detect S-adenosylmethionine or acetylcholine in either human or mouse sera. This inability may have been due to the short half-lives of these metabolites: S-adenosylmethionine is eliminated from blood within a few hours by several mechanisms including metabolic degradation, excretion in the urine, and transfer to organs[43–45], and acetylcholine has a half-life of about 2 min in blood and elsewhere in the body owing to its rapid hydrolysis by cholinesterase[46]. Furthermore, although the administration of *B. wexlerae* increased the fecal ʟ-ornithine content of HFD-fed mice (Supplementary Fig. 23A), serum levels of ʟ-ornithine were comparable among *B. wexlerae*-untreated CD-fed and HFD-fed mice and *B. wexlerae*-receiving HFD-fed mice (Supplementary Fig. 23B). Orally supplemented ʟ-ornithine is converted to several metabolites through host amino acid and urea metabolic pathways[47,48], collectively suggesting that exogenous ʟ-ornithine appears to be

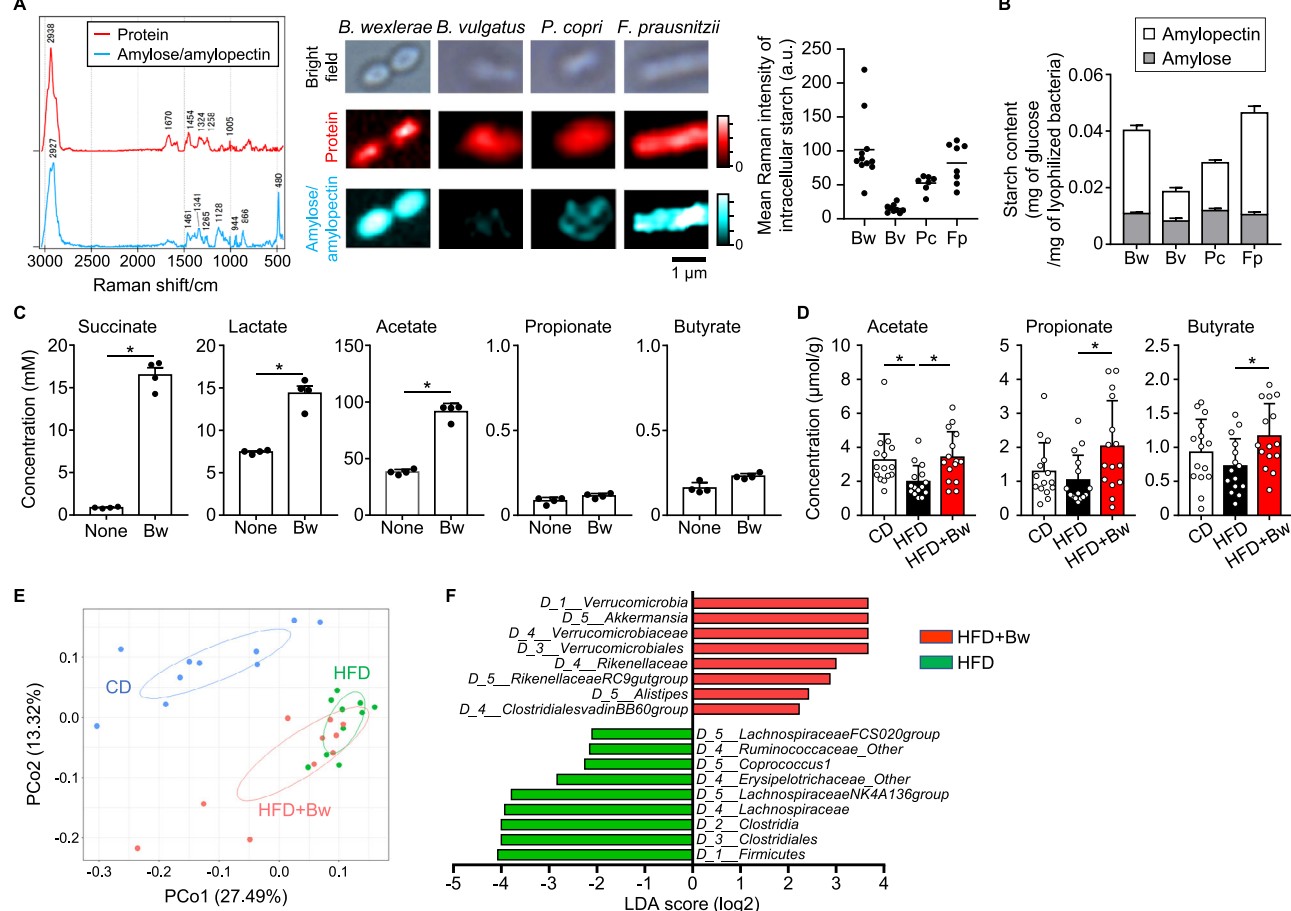

**Fig. 5 | Administration of *B. wexlerae* altered the intestinal environment, including gut bacterial composition and fecal short-chain fatty acid (SCFA) content, in mice. A** Raman spectroscopic analysis. Raman shift signals specific for protein and starch were plotted on bacterial cells of *B. wexlerae* (Bw) and major intestinal bacteria including *Bacteroide vulgatus* (Bv), *Prevotella copri* (Pc), and *Faecalibacterium prausnitzii* (Fp). a.u., arbitrary units. **B** Starch (amylose and amylopectin) contents in *B. wexlerae* (Bw) and major intestinal bacteria, including *B. vulgatus* (Bv), *P. copri* (Pc), and *F. prausnitzii* (Fp). Data are representative of two independent experiments (*n* = 4 biologically independent samples, mean ± 1 SD). **C** The concentrations of succinate, lactate, acetate, propionate, propionate, and butyrate in fresh medium (none) and *B. wexlerae* culture supernatant (Bw). Data are representative of two independent experiments (*n* = 4, mean ± 1 SD). * *P* = 0.0286 (two-tailed Mann–Whitney *U* test). **D** SCFA content in fecal samples from mice.

Mice were maintained on CD or HFD for 8 weeks with or without oral administration of *B. wexlerae* three times each week, after which fecal SCFAs were measured by HPLC. Data are combined from three independent experiments (*n* = 15, mean ± 1 SD). *P < 0.05 (one-way ANOVA). **E** Principal coordinate analysis (PCoA) of fecal bacterial composition in mice according to the Bray–Curtis distance at genus level. Mice were maintained on CD or HFD for 8 weeks with or without oral administration of *B. wexlerae* three times each week, after which fecal bacterial composition was analyzed by 16S amplicon sequencing. Data are combined from two independent experiments (*n* = 10). (F) Differences in bacterial taxonomy were ranked according to the linear discriminant analysis (LDA) effect size between HFD-fed mice and HFD-fed mice supplemented with *B. wexlerae*. CD, CD-fed mice; HFD, HFD-fed mice, HFD + Bw, HFD-fed mice supplemented with *B. wexlerae*.

---

utilized for host metabolism, rather than circulating in its native form. In humans, no relationship between *Blautia* abundance and serum or fecal ʟ-ornithine levels has yet been identified, probably because they are affected by several factors including dietary habits, age, and urea cycle activity[42,49]. Despite the need for further studies on these metabolites that address their kinetics and how they affect distinct organs such as adipose tissue from the intestine, it remains clear that *B. wexlerae* produces biologically active metabolites, which may account in part for the health-beneficial effects of this organism.

## Administration of *B. wexlerae* to mice modified their intestinal environment

To identify additional *B. wexlerae* candidate molecules for the control of obesity and diabetes other than those associated with amino acid metabolism, we performed comprehensive Raman spectroscopic analysis to detect multiple molecules simultaneously without labeling[50,51]. As a control signal, the Raman spectra for total protein were identical among *B. wexlerae*, *B. vulgatus*, *P. copri*, and *F.*

*prausnitzii* (Fig. 5A). In contrast, the spectrum characteristic of *B. wexlerae* was extracted by multivariate curve resolution analysis. This spectrum has a sharp peak at 480 cm⁻¹ and several Raman bands attributed to carbohydrates are observed in the 800–1500 cm⁻¹ region. These all bands are assigned to amylose and amylopectin[52]. The corresponding intensity images indicate that these polysaccharides are highly accumulated in *B. wexlerae* and *F. prausnitzii* cells, but only slightly in *B. vulgatus* and *P. copri* cells (Fig. 5A). Plotting the intracellular mean intensity also shows the same tendency with variation among single cells, suggesting that some *B. wexlerae* cells have a large amount of accumulation.

Starch consists of amylose and amylopectin, which are linear and branched, respectively, polymers of glucose units. The amylose content was similar among *B. wexlerae*, *B. vulgatus*, *P. copri*, and *F. prausnitzii* (Fig. 5B). In contrast, amylopectin was more abundant in *B. wexlerae* than in *B. vulgatus* and *P. copri* but similar to that in *F. prausnitzii*, the carbohydrate preferences of which have been predicted through in silico techniques[53] (Fig. 5B). Consistent with these findings,

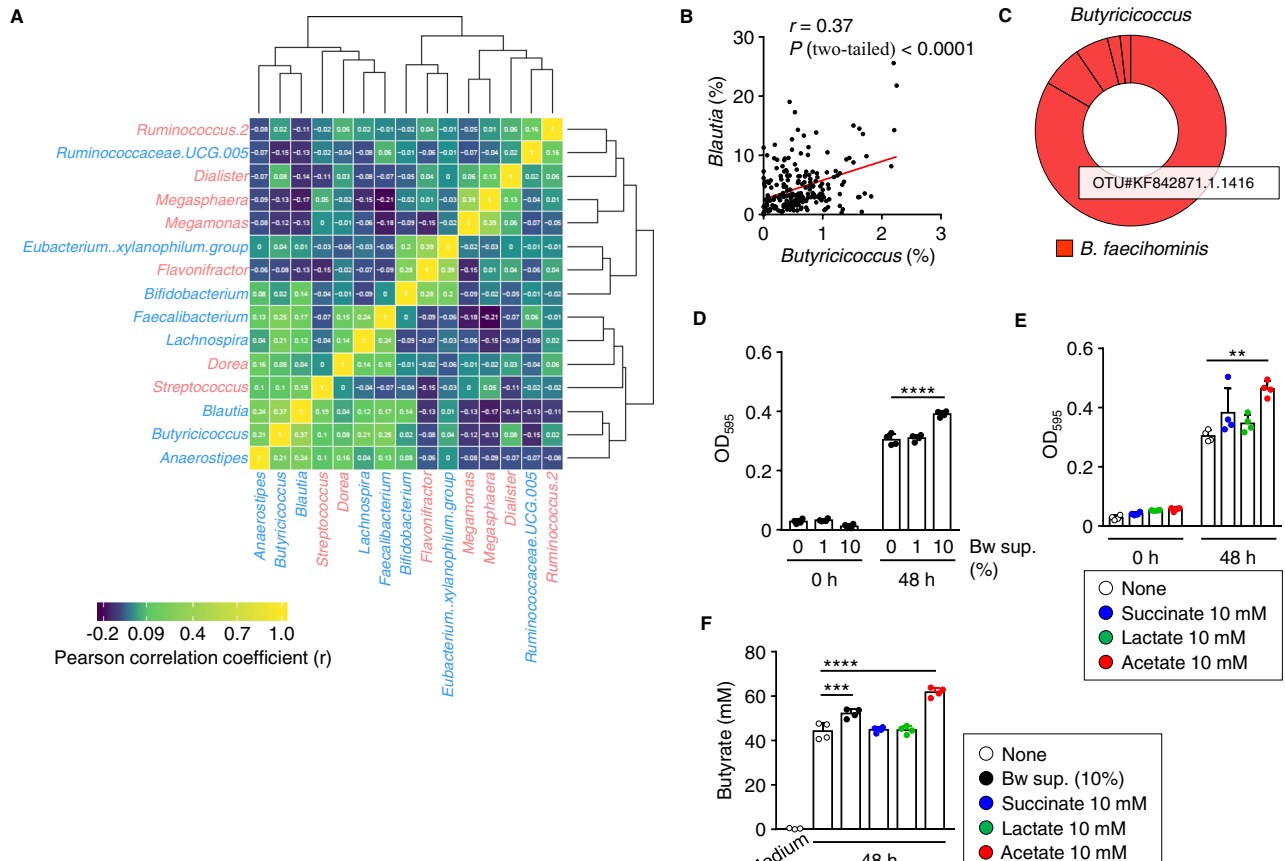

**Fig. 6 | Interaction between *B. wexlerae* and butyrate-producing bacteria.**
**A** Heatmap showing correlated relationship among human intestinal bacteria (*n* = 217). Red and blue fonts indicate bacterial genera from Fig. 1 that were positively or inversely, respectively, related to BMI/T2DM. **B** Positive correlation between *Blautia* and *Butyricicoccus* in human fecal samples (*n* = 217) (Pearson, two-tailed *P* value). **C** Estimation of *Butyricicoccus* species according to BlastN analysis of representative OTU sequences. **D** The absorbance of *Butyricicoccus faecihominis*-cultured medium in which the organisms were grown in the absence or presence of *B. wexlerae*-cultured medium at a final concentration of 1% or 10% (*n* = 4,

mean ± 1 SD). ****$P < 0.0001$ (one-way ANOVA). **E** The absorbance of *B. faecihominis*-cultured medium in which the organisms were grown in the absence (none) or presence of 10 mM succinate, 10 mM lactate, or 10 mM acetate (*n* = 4, mean ± 1 SD). **$P = 0.0019$ (one-way ANOVA). **F** The concentration of butyrate in *B. faecihominis*-cultured medium in which the organisms were grown in the absence (none) or presence of *B. wexlerae* culture supernatant at 10%, 10 mM succinate, 10 mM lactate, or 10 mM acetate (*n* = 3–4, mean ± 1 SD). ***$P = 0.0010$; ****$P < 0.0001$ (one-way ANOVA). Data are representative of two independent experiments (D–F).

genomic information (GenBank accession no. CZAW01000000) indicates that *B. wexlerae* harbors the necessary metabolic enzymes for the de novo synthesis of starch from glucose and sucrose (Supplementary Fig. 24). Therefore, the high concentration of starch observed in *B. wexlerae* was likely due to its high amylopectin content.

Bacterial metabolization of amylose and amylopectin results in the generation of several SCFAs[54], and genomic information (GenBank accession no. CZAW01000000) indicates that *B. wexlerae* conserves various enzymes necessary for the production of succinate, lactate, and acetate but lacks production pathways for propionate and butyrate (Fig. S25). Indeed, succinate, lactate, and acetate contents were increased in *B. wexlerae*-cultured medium compared with fresh medium, but propionate and butyrate levels were similar between cultured and fresh media (Fig. 5C). In addition, the concentration of succinate in the supernatant from *B. wexlerae* cultures was higher than that from *P. copri* and *F. prausnitzii* and similar to that from *B. vulgatus* (Figs. 5C and S26). Furthermore, lactate and acetate were more abundant in the supernatants from cultures of *B. wexlerae* than from other bacteria (Fig. 5C, Supplementary Fig. 26). Together, these findings indicate that *B. wexlerae* utilizes abundantly stored starch (i.e., amylopectin) and produces increased levels of succinate, lactate, and acetate.

Because cross-feeding, such as the utilization of acetate by butyrate-producing bacteria, is an important metabolic interaction

among intestinal bacteria[54,55], we next examined the concentrations of various SCFAs in fecal samples from mice. In the results, propionate and butyrate were somewhat decreased and acetate was markedly reduced in feces from HFD-fed mice compared with CD-fed mice (Fig. 5D). In contrast, the administration of *B. wexlerae* to HFD-fed mice increased fecal levels of acetate, propionate, and butyrate (Fig. 5D). None of the groups of mice yielded detectable fecal levels of succinate or lactate. These findings collectively suggest that *B. wexlerae* does not produce propionate and butyrate but supplies substrates such as succinate, lactate, and acetate to other gut commensal bacteria, consequently leading to the increased levels of propionate and butyrate in feces.

Furthermore, to consider the influence of *B. wexlerae* on the composition of other commensal bacteria in the gut, we used 16S rRNA sequencing analysis to examine fecal bacterial composition in mice. Consistent with previous reports[54], principal coordinate analysis (PCoA) revealed that gut bacterial composition differed between HFD- and CD-fed mice (Fig. 5E). When HFD-fed mice were treated with *B. wexlerae*, their bacterial composition was closer to (but still different from) that of HFD-fed mice than CD-fed mice (*P* < 0.01, HFD vs HFD + Bw) (Fig. 5E).

The *Blautia* abundance was reduced in HFD-fed mice compared with CD-fed mice (Supplementary Fig. 27A), consistent with its low

representation during human obesity (Supplementary Fig. 1A). The administration of *B. wexlerae* did not influence its own abundance (Supplementary Fig. 27A), as confirmed through qPCR analysis (Supplementary Fig. 27B). These results suggest that orally administered *B. wexlerae* cannot colonize the intestine of mice under the experimental conditions we used, probably because the niche already is occupied by intestinal commensal bacteria including the existing *Blautia*.

Linear discriminant analysis effect size[56] provided a ranked list of intestinal bacteria that differed between nonsupplemented HFD-fed mice and HFD-fed mice given *B. wexlerae* (Fig. 5F). At the genus-level, *Lachnospiraceae NK4A136 group* was increased in HFD-fed mice compared with HFD-fed mice given *B. wexlerae*, which were similar to CD-fed mice in regard to these organisms (Supplementary Fig. 27C). In contrast, the proportions of *Rikenellaceae RC9 gut group* and *Alistipes* were decreased in HFD-fed mice compared with CD-fed mice; the administration of *B. wexlerae* to HFD-fed mice increased these organisms to similar levels as in CD-fed mice (Supplementary Fig. 27C). Furthermore, the proportion of *Akkermansia* was increased in HFD-fed mice compared with CD-fed mice and increased further upon administration of *B. wexlerae* (Supplementary Fig. 27C). Therefore, the administration of *B. wexlerae* likely changed the gut bacterial composition to increase the SCFA production capacity by increasing *Rikenellaceae RC9 gut group*, *Alistipes*, and *Akkermansia*, all of which produce propionate, butyrate, or both[57–59]. *Akkermansia* has well established metabolic benefits;[60,61] its increased proportion in *B. wexlerae*-treated HFD-fed mice perhaps supports the beneficial effects of *B. wexlerae* and suggests that the increase in *Akkermansia* is mediated by metabolites produced from *B. wexlerae*. Indeed, lactate and acetate promote the growth of *A. muciniphila*[62,63], and protracted administration of lactate- or acetate-producing bacteria, such as *Lactobacillus* and *Bifidobacterium*, also increases *Akkermansia* abundance[63,64].

These results from our mouse experiments provided feedback for our human data analysis. Among the 15 bacterial genera that were related to obesity and T2DM in our Japanese cohort (Fig. 1), the *Blautia* genus was positively correlated with several genera, especially *Butyricicoccus*, which is a butyrate-producing bacterium[59] (Fig. 6A, B). This relationship suggests that *Blautia* might influence features of the intestinal environment, such as SCFA production, in collaboration with other intestinal bacteria. To address this hypothesis, we examined the effects of *B. wexlerae*-derived metabolites on the growth and SCFA production of *Butyricicoccus faecihominis*, a dominant species of the *Butyricicoccus* genus (Fig. 6C).

*B. faecihominis* was grown anaerobically and monitored at OD$_{595}$. The absorbance of *B. faecihominis* at 48 h was increased dose-dependently according to supplementation with *B. wexlerae*-cultured supernatant (Fig. 6D). Consistent with this finding, supplementation with acetate, a product of *B. wexlerae*, likewise increased the growth of *B. faecihominis* (Fig. 6E). In addition, supplementation with *B. wexlerae*-cultured supernatant or acetate increased the amount of butyrate produced by *B. faecihominis* (Fig. 6F). These findings indicate a metabolic interaction between *B. wexlerae* and *B. faecihominis*, a butyrate-producing intestinal bacterium; this interaction may, in turn, improve various aspects of the intestinal environment, such as butyrate production.

## Discussion

By combining human cohort data and animal experiments, we have demonstrated that *B. wexlerae* has potent effects for reducing obesity and T2DM. Our human cohort studies showed that intestinal *B. wexlerae* was associated with a decreased risk of obesity and diabetes in Japanese adults. Regarding a possible underlying mechanism, *B. wexlerae* contains several amino acid metabolites (i.e., S-adenosylmethionine, acetylcholine, L-ornithine) that confer anti-adipogenesis and anti-inflammatory properties to adipocytes. In addition, *B. wexlerae* contains a high amount of starch (i.e., amylopectin) and produces

carbohydrate metabolites (i.e., succinate, lactate, acetate) to consequently modify the gut environment, including the bacterial and SCFA composition of the gut microbiota. Thus, by initiating these complex changes, *B. wexlerae* likely offers various, numerous, and diverse benefits for health maintenance.

Accumulating evidence reveals several human intestinal bacteria associated with obesity and T2DM. For example, the genera of *Bifidobacterium*, *Faecalibacterium*, *Roseburia*, and *Akkermansia* are inversely associated with obesity and T2DM, are beneficial to human health, and may be useful as probiotics[61,65,66]. Indeed, in our Japanese cohort, *Bifidobacterium* and *Faecalibacterium* emerged as beneficial bacterial genera inversely associated with BMI or T2DM. Although reported less frequently than *Bifidobacterium* and *Faecalibacterium*, the genera *Megasphaera* and *Ruminococcus* are both associated with obesity or T2DM[16,67], consistent with our results.

Several previous human studies have indicated the involvement of intestinal *Blautia* in obesity and T2DM, but whether *Blautia* contributes to aggravation or amelioration of these disorders is unclear[16,68]. For example, in Russian subjects, disorders of glucose metabolism, such as high fasting glucose levels, were associated with increased prevalence of *Blautia*[69]. In contrast, a European study found no significant correlation between *Blautia* and obesity or T2DM[7]. However, a Spanish study of the childhood microbiome and its association with the metabolic complications underlying obesity revealed that *Blautia* species, including *B. wexlerae*, were associated with normal-weight children and negatively correlated with fecal inflammatory cytokines such as TNF-α[70]. In addition, a previous Japanese cohort study reported that *Blautia* was significantly and inversely associated with visceral fat area, regardless of sex[29]. In view of these apparent inconsistencies, it is important to remember that regional variation in human gut microbial composition occurs. Indeed, the genus *Blautia* is more abundant in the gut microbiota of Japanese people than those of other countries[18], and studies involving Japanese cohorts consistently indicate beneficial effects of *Blautia*. Furthermore, at the species level, the *Blautia* genus includes several OTUs; OTU GQ448486.1.1387 of *B. wexlerae* was dominant in our Japanese cohort, but we wonder which species are abundant in other geographic regions. Given that our current study identified multiple potentially health-promoting molecules in *B. wexlerae*, the inconsistent findings among other studies might be clarified by examining the differences among bacterial strains in light of their ability to produce beneficial metabolites.

In this regard, we identified several products of *B. wexlerae* amino acid metabolism that can contribute to the control of obesity and diabetes. Acetylcholine is a neurotransmitter and an agonist for the alpha-7 nicotinic acetylcholine receptor, which is expressed on adipocytes and is a potent therapeutic target for inflammatory diseases, including T2DM and obesity[71–74]. In addition, numerous central neurotransmitters are present in the gastrointestinal tract, where their local effects include modulating gut motility and cell signaling[75]. The absence of a gut microbiota is associated with significant reduction in the intestinal levels of these neurotransmitters, suggesting that gut microbes are important producers of neurotransmitters[75]. Although *Lactobacillus plantarum*, which was isolated from a fermented food (sauerkraut), is known to synthesize acetylcholine[76], no human intestinal acetylcholine-producing bacteria had been identified previously. We here revealed that the human intestinal bacteria *B. wexlerae* produces acetylcholine. An interesting potential subject for future study is the acetylcholine synthetic pathway of *B. wexlerae*.

Another *B. wexlerae*-derived amino acid metabolite that we identified is L-ornithine; its beneficial effects include roles in detoxifying ammonia, as an intermediate in the urea cycle, and in modulating host lipid metabolism[48,77]. Intriguingly, even though L-ornithine is a nutrient necessary for health maintenance, it is difficult to obtain sufficient L-ornithine from ordinary diets; most foods have only scant

levels of L-ornithine[48]. In addition, a lack of intestinal bacteria reportedly hampers hepatic L-ornithine homeostasis in mice[78]. These previous and our current findings together support the importance of intestinal bacteria, including *Blautia*, in the supply of L-ornithine for health maintenance.

S-adenosylmethionine, a third *B. wexlerae*-derived metabolite, acts as a methyl donor for biologic methylation and helps to prevent diabetes, obesity, and inflammation[79,80]. In addition, S-adenosylmethionine is applied in the treatment of depression, osteoarthritis, and liver diseases[81]. Because S-adenosylmethionine is therefore an important substance in health maintenance, this metabolite and the microbes that produce it have garnered attention in regard to their applications in the pharmaceutical and food industries[82]. However, the mechanisms through which intestinal bacteria produce and supply S-adenosylmethionine are not well understood[83,84]. Our findings that *B. wexlerae*, an intestinal bacterium, produces S-adenosylmethionine, L-ornithine, and acetylcholine collectively point to the importance of amino acid metabolism by gut microbiota both in the development of metabolic disorders such as obesity and T2DM and in strategies for their prevention and treatment.

Starch is one of the main forms of carbohydrate in the diet[85]. Digestible starches are broken down by enzymes in the small intestine and then absorbed. In contrast, indigestible resistant starches pass into the colon, where they induce physiologic consequences including improvements in glucose tolerance and insulin response via modification of gut environment such as SCFA condition[85]. The indigestible components of plant foods, such as cell wall materials, are rich sources of dietary fiber. In the current study, we revealed that *B. wexlerae* contains high concentrations of starch, especially amylopectin. We think that when a probiotic containing *B. wexlerae* is administered, the starch concentrated in *B. wexlerae* acts as 'bacterial fiber' that affects the gut environment, much like the dietary fiber provided from plants. Indeed, oral supplementation with *B. wexlerae* influenced gut microbial composition and increased fecal propionate and butyrate levels in mice, even though *B. wexlerae* itself cannot produce these SCFAs. In addition to providing starch, *B. wexlerae* generates succinate, lactate, and acetate through carbohydrate metabolism. Acetate is utilized by butyrate-producing bacteria to produce butyrate[54,55,86]. Indeed, our current study revealed the cooperative effects of *B. wexlerae* on butyrate production by *Butyricicoccus*. Intestinal bacteria also utilize succinate and lactate, converting them into propionate or butyrate[87]. In this regard, *B. faecihominis* appeared to produce propionate in addition to butyrate, albeit at lower levels, and supplementation with succinate enhanced the organism's propionate production. In terms of bacterial cross-feeding, *Bifidobacterium* is a well-known provider of acetate and lactate—but not succinate[87,88]. *B. wexlerae*, which generates succinate in addition to acetate and lactate, seems to be a key microbial player in driving metabolic conditions in the gut. Overall, carbohydrate metabolism in *B. wexlerae* appears to improve the gut environment, leading to amelioration of metabolic disorders.

Because of the observational nature of our human study, whether *B. wexlerae* in fact reduces obesity and diabetes in humans is unclear, so follow-up interventional and other studies are needed. Whether our findings apply globally needs to be ascertained, given differences in lifestyle, host metabolism, and the resident gut microbiota among humans worldwide. In addition, our diet-induced mouse model qualifies as an insulin-resistant and pre-diabetic model, and the *Blautia*-associated anti-diabetic effects we revealed should be verified using T2DM models such as the streptozocin-induced model and various genetically deficient mouse strains. In this diet-induced model, male mice are generally used because they are more susceptible to diet-induced weight gain and diabetic pathology[89], but it is also necessary to examine the sexual differences of the effects of *B. wexlerae* in the future study including human. As done in previous studies[90,91], we fasted our mice for 16 h; however, this might have been excessive and thus induced metabolic changes that occur during starvation. Furthermore, although we focused on the metabolism of adipose tissue as a possible mechanism of the control of obesity and diabetes, liver has an even greater role than adipose tissue in energy metabolism, such that more comprehensive studies including assessments using metabolic cages are needed. Finally, even when they belong to the same species, different microbial strains differ genetically and perhaps functionally[92–94]. Our findings were obtained by using a type strain, and the actual strains involved in the clinical effect should be isolated and subjected to single-cell analysis and other evaluation. Despite these limitations, our findings provide important insights into the contributions of the gut microbiota to overall health.

In summary, we identified *B. wexlerae* as a potent beneficial human intestinal bacterium with anti-inflammatory properties and the ability to modify the host gut environment and lipid metabolism. Regarding the mechanisms underlying these benefits, we elucidated characteristics of *B. wexlerae*'s amino acid and carbohydrate pathways, in which multiple metabolites appear to act cooperatively to ameliorate metabolic disorders including obesity and diabetes. The discovery of a metabolic role of the gut microbiota in the pathophysiology of obesity and T2DM reveals opportunities for the development of preventive methods (e.g., probiotics) and therapeutic approaches for metabolic disorders.

## Methods

### Collection of human samples

All experiments were approved by the Ethics Committee of the National Institutes of Biomedical Innovation, Health, and Nutrition and were conducted in accordance with their guidelines (approval numbers: 177-08 and Kenei78-06). Informed consent was obtained from all participants.

For the discovery cohort, diabetic patients were recruited at Shinnanyo Hospital, Shunan City, Yamaguchi, Japan (Table S1); non-diabetic adult volunteers (control subjects; e.g., staff members of city offices and chamber of commerce) were recruited at health examination sites in surrounding communities (Table S1). From the 242 total participants possible, we excluded those who had received antibiotics within the previous 2 weeks (18 subjects), had traveled overseas within the previous month (two subjects), or had gastrointestinal disease (5 subjects), thus leaving a study population of 217 participants comprising 147 nondiabetic subjects, 45 patients with type 2 diabetes, and 25 patients with type 1 diabetes.

For a validation cohort, we recruited adult volunteers from among the city office staff of Minamiuonuma City, Niigata, Japan, at health examination sites (Table S8). From the 219 total participants possible, we excluded 24 who had received antibiotics within the previous 2 weeks (ten subjects), had traveled overseas within the previous month (six subjects), or had gastrointestinal disease (eight subjects), thus leaving a study population of 195 participants; all of these volunteers were nondiabetic.

Fecal samples were collected and placed in guanidine thiocyanate solution (TechnoSuruga Laboratory Co., Ltd, Shizuoka, Japan) as previously described[95], which allows the sample to be stored at room temperature. The participants collected their stool sample at home without any restrictions, such as fasting, and submitted it to the hospital (diabetic patients) or health examination site (control subjects) within 5 days. Blood samples were collected at the hospital and health examination sites. Physical measurements, including body weight, height, blood glucose, and HbA1c, and disease information were obtained through health examinations and from medical records.

### 16S rRNA gene amplicon sequencing analysis

DNA was extracted from human fecal samples in guanidine thiocyanate solution by using the bead beating method and an automatic nucleic

acid extraction system (Gene Prep Star PI-80X, Kurabo Industries, Ltd, Osaka, Japan) as previously described[95]. DNA was extracted from mouse fecal samples through a slight modification of this method. Briefly, a mouse fecal sample was placed in a 2-ml vial (Wakenbtech Co., Ltd, Tokyo, Japan) containing 0.5 ml of lysis buffer (No. 10, Kurabo Industries, Ltd) and 0.5 g of 0.1-mm glass beads. The mixture was mechanically disrupted by bead beating by using a Cell Destroyer PS1000 (Bio Medical Science, Tokyo, Japan) at 4260 rpm for 50 s at room temperature. After centrifugation at $13,000 \times g$ for 5 min at room temperature, DNA was extracted from 0.2 ml of the supernatant by using a Gene Prep Star PI-80X device (Kurabo Industries, Ltd).

The 16S rRNA gene amplicon in human and mouse fecal DNA was sequenced as previously described[95]. The V3–V4 region of the 16S rRNA gene was amplified from the fecal DNA samples by using the following primers: forward, 5′- TCGTCGGCAGCGTCAGATGTGTATAAGCGACAG CCTACGGGNGGCWGCAG-3′; and reverse, 5′-GTCTCGTGGGCTCGGA GATGTGTATAAGAGACAGGACTACHVGGGTATCTAATCC-3′[96]. A DNA library was prepared by using a Nextera kit Set A (Illumina, San Diego, California, USA), and 16S rRNA gene sequencing was performed by using MiSeq (Illumina) in accordance with the manufacturer's instructions. The sequencing results were analyzed using the Quantitative Insights Into Microbial Ecology (QIIME) software package[97] and QIIME Analysis Automating Script (Auto-q) (https://doi.org/10.5281/zenodo.1439555) as previously described[98]. Open-reference operational taxonomic unit (OTU) picking and taxonomy classification were performed based on sequence similarity (>97%) by using UCLUST software[99] with the SILVA v128 reference sequence[100].

## Bacterial strains and culture

*B. wexlerae* (JCM 17041), *B. vulgatus* (JCM 5826), *P. copri* (JCM 13464), *F. prausnitzii* (JCM 31915), and *B. faecihominis* (JCM 31056) were provided by the RIKEN BRC through the National BioResource Project of the MEXT/AMED, Japan. All bacterial strains were cultured anaerobically at 37 °C by using an anaerobic chamber (Bactron 300, Toei Kaisha, Ltd, Tokyo, Japan). For oral administration into mice, *B. wexlerae* was cultured anaerobically in reinforced clostridial medium (BD Difco, Franklin Lakes, NJ, USA) at 37 °C for 48 h until $OD_{600} = 1.0–1.5$. Cultures were stored as 0.5-ml aliquots at –80 °C until use for oral administration into mice. For measurement of starch, SCFAs, and metabolites and Raman spectroscopic analysis, *B. wexlerae*, *B. vulgatus*, *P. copri*, and *F. prausnitzii* were cultured in clostridial reinforced medium (BD Difco) at 37 °C for 48 h; at inoculation, $OD_{600}$ was approximately 0.05 for all strains (*B. wexlerae*, 1.0; *B. vulgatus*, 1.1; *P. copri*, 0.8; and *F. prausnitzii*, 0.9). For the cross-feeding assay, *B. faecihominis* was cultured anaerobically at 37 °C in GAM broth (Nissui Pharmaceutical Co., Ltd, Tokyo, Japan) supplemented without or with *B. wexlerae* culture supernatant, succinate acid, sodium lactate, or sodium acetate; the pH of the medium was adjusted to 7.0.

## Mice and high-fat diet-induced obesity and diabetes

Male C57BL/6 mice (age, 4 weeks) were purchased from Japan SLC (Shizuoka, Japan). Each group of five mice was maintained in a single cage in the specific pathogen-free animal facility and were fed a standard diet (Oriental Yeast, Tokyo, Japan; AIN-93M) for 2 weeks, followed by a high-fat diet composed of chemically defined materials (Oriental Yeast; AIN-93G) for 10 weeks; accordingly, this mouse model qualifies as an insulin-resistant and pre-diabetic model. During the duration of the experiments, mice had free access to food and water and received 0.5 ml of bacterial solution (i.e., $5 \times 10^8$ CFU) or fresh medium (as a vehicle control) by oral gavage three times each week, and the weights of each mouse and the total amount of diet consumed by each cage of mice were calculated every week. At 8 weeks, blood and fecal samples were collected for calculation of HOMA-IR and for measurement of the gut microbiome and SCFAs, respectively, and IPGTT was performed. At 9 weeks, a blood sample was collected for monitoring serum insulin

concentration over time. At 10 weeks, mice were euthanized by cervical dislocation, after which tissues were collected by using surgical scissors. All experiments involving mice were approved by the Animal Care and Use Committee of the National Institutes of Biomedical Innovation, Health, and Nutrition (approval no. DS27-48R10) and were conducted in accordance with their guidelines.

## Intraperitoneal glucose tolerance testing (IPGTT)

IPGTT was performed as described previously[90] with modification. Briefly, mice were fasted overnight (16 h) and then intraperitoneally injected with D-(+)-glucose (2 g/kg body weight; 20% solution, Nacalai Tesque, Kyoto, Japan). To measure the blood glucose level, blood was obtained from the tail vein by cutting with a single-edged blade (Feather, Osaka, Japan) and analyzed on a ONE TOUCH Ultra Vue (LifeScan Japan, Tokyo, Japan) before and after glucose injection at the indicated time points. Blood insulin levels were measured by using an LBIS Mouse Insulin ELISA kit (Wako Pure Chemicals, Osaka, Japan) in accordance with the manufacturer's instructions. Samples unsuitable for analysis, such as hemolyzed blood, were excluded.

## Histological analysis

Frozen tissue was analyzed histologically as described previously with minor modification[101]. Briefly, tissue samples were washed with PBS (Nacalai Tesque) on ice and frozen in Tissue-TeK OCT compound (Sakura Finetek, Tokyo, Japan) in liquid nitrogen. Frozen tissue sections (thickness, 6 μm) were prepared by using a cryostat (model CM3050 S, Leica, Wetzlar, Germany) and were fixed for 30 min at 4 °C in prechilled 95% ethanol (Nacalai Tesque) followed by 1 min at room temperature in prechilled 100% acetone (Nacalai Tesque).

For immunohistological analysis, tissue sections were washed with PBS for 10 min and then blocked in 2% (vol/vol) newborn calf serum in PBS for 30 min at room temperature in an incubation chamber (Cosmo Bio, Tokyo, Japan). Tissue sections were incubated with purified anti-F4/80 monoclonal antibody (1:100; catalog no. 123102, Biolegend, San Diego, CA, USA) and BODIPY493/503 (1:1000; catalog no. D3922, Molecular Probes, Eugene, Oregon, USA) in 2% (vol/vol) newborn calf serum in PBS for 16 h at 4 °C in the incubation chamber, washed once for 5 min each in 0.1% (vol/vol) Tween-20 (Nacalai Tesque) in PBS and in PBS only, and then stained with Cy3-anti-rat IgG (1:200; catalog no. 712-165-153, Jackson Immuno Research Laboratories, West Grove, Pennsylvania, USA) in 2% (vol/vol) newborn calf serum in PBS for 30 min at room temperature in the incubation chamber. To visualize nuclei, tissue sections then were washed twice (5 min each) with PBS and stained with DAPI (1 μM, AAT Bioquest, Sunnyvale, CA, USA) for 10 min at room temperature in the incubation chamber. Finally, tissue sections were washed twice with PBS, mounted in Fluoromount (Diagnostic BioSystems, Pleasanton, CA, USA), and examined under a fluorescence microscope (model BZ-X810, Keyence, Osaka, Japan); areas of fluorescence were calculated by using the software provided with the microscope (version 1.1.2.4, Keyence).

## Cell preparation from epididymal adipose tissue

Epididymal adipose tissue (eAT) was minced into Krebs–Ringer bicarbonate HEPES (KRBH) buffer (120 mM NaCl, 4 mM $KH_2PO_4$, 1 mM $MgSO_4$, 1 mM $CaCl_2$, 10 mM $NaHCO_3$, 30 mM HEPES, 20 μM adenosine, and 4% [wt/vol] BSA) by using scissors, centrifuged to remove blood cells, and then incubated in 2.67 mg/ml collagenase in KRBH buffer for 45 min at 37 °C with stirring. After the samples were centrifuged, we collected the stromal vascular fraction (i.e., the pellet, which contains immune cells) and mature adipocyte fraction (MAF; i.e., the supernatant) for flow cytometric analysis and RT-qPCR testing.

## Flow cytometric analysis

Flow cytometry was performed as described previously, with slight modification[91,102]. Cells in the stromal vascular fraction of eAT were

stained with an anti-CD16/32 monoclonal antibody (TruStain fcX; Biolegend) to avoid non-specific staining and 7-AAD (Biolegend) to detect dead cells. The cells were further stained with the fluorescently labeled antibodies BV421–anti-CD45 (Biolegend, clone 30-F11), PE–anti-I-A$^b$ MHC class II (Biolegend, clone AF6-120.1), FITC-anti-CD206 (Biolegend, clone C068C2), PE-Cy7–anti-F4/80 (Biolegend, clone BM8), and APC-Cy7–anti-CD11b (Biolegend, clone M1/70). Samples were analyzed by using BD FACSAria III (BD Biosciences), and data were analyzed by using FlowJo 9.9 (Tree Star, Ashland, Oregon, USA). The CD45$^+$CD11b$^+$F4/80$^+$ cells were defined as the macrophage population[91]. Among macrophages, the MHC II$^{+/high}$CD206$^{–/low}$ cells and MHC II$^+$CD206$^{high}$ cells were defined as the M1- and M2-like macrophage populations, respectively.

### Reverse transcription and quantitative PCR (RT-qPCR) analysis

RT-qPCR analysis were performed as described previously[101] with minor modification. Total RNA was isolated from purified or cultured cells by using Sepasol (Nacalai Tesque) and chloroform (Nacalai Tesque), precipitated by using 2-propanol (Nacalai Tesque), and washed with 75% (vol/vol) ethanol (Nacalai Tesque). RNA samples were incubated with DNase I (Invitrogen, Carlsbad, California, USA) to remove contaminating genomic DNA and then reverse-transcribed into cDNA (Superscript III reverse transcriptase, VIRO cDNA Synthesis Kit; Invitrogen).

Quantitative PCR analysis was performed by using a LightCycler 480 II (Roche, Basel, Switzerland) with FastStart Essential DNA Probes Master (Roche) or SYBR Green I Master reagents (Roche). Primer sequences were: *Tnfα* sense, 5′-CTGTAGCCCACGTCGTAGC-3′; *Tnfα* anti-sense, 5′-TTGAGATCCATGCCGTTG-3′; *S100a8* sense, 5′-TCCTT GCGATGGTGATAAAA-3′; *S100a8* anti-sense, 5′-GGCCAGAAGCTC TGCTACTC-3′; *Pparγ* sense, 5′-GAAAGACAACGGACAAATCACC-3′; *Pparγ* anti-sense, 5′-GGGGGTGATATGTTTGAACTTG-3′; *Nrf1* sense, 5′-GCTCTCTGAGACGCTGCTTT-3′; *Nrf1* anti-sense, 5′-GTGTTCAGTT TGGGTCACTCC-3′; *Actb* sense, 5′-AAGGCCAACCGTGAAAAGAT-3′; and *Actb* anti-sense, 5′-GTGGTACGACCAGAGGCATAC-3′.

### 3T3L1 adipocytes

The 3T3L1 cell line (JCRB 9014) was purchased from the Japanese Collection of Research Bioresources (JCRB) cell bank (Osaka, Japan). Differentiation of 3T3L1 adipocytes was performed as previously reported with modifications[103]. 3T3L1 cells were seeded at $3.5 × 10^4$ cells per well in 12-well plates and incubated overnight (16 h) in DMEM (Nacalai Tesque) at 37 °C under 5% CO$_2$ supplemented with 10% (vol/vol) newborn calf serum. The following day, the medium on the cells was changed to DMEM (Nacalai Tesque) supplemented with 10% (vol/vol) fetal bovine serum, and cells were cultured for 3 d; cells were then cultured in adipocyte differentiation medium (0.5 mM isobutyl-methylxanthine [Sigma-Aldrich, St. Louis, Missouri, USA], 1 μM dexamethasone [Sigma-Aldrich], 10 μg/ml insulin [Sigma-Aldrich] in DMEM [Nacalai Tesque] supplemented with 10% [vol/vol] fetal bovine serum for 2 d and finally in DMEM (Nacalai Tesque) supplemented with 10% (vol/vol) fetal bovine serum for 6 d. During the terms, 1 or 10% *B. wexlerae*-cultured medium or uncultured fresh medium (reinforced clostridial medium [BD Difco]) was added to the 3T3L1 culture medium.

### Mitochondrial mass analysis by flow cytometry using Mitogreen

For mitochondrial staining, 3T3L1 adipocytes were incubated with 100 nM Mitogreen (Takara Bio, Shiga, Japan) in PBS supplemented with 2% (vol/vol) newborn calf serum for 15 min at 37 °C under 5% CO$_2$. Stained cells were washed with PBS, treated with trypsin, and then suspended in PBS supplemented with 2% (vol/vol) newborn calf serum. Samples were analyzed by using MACSQuant (Miltenyi Biotec, Bergisch Gladbach, Germany), and data were analyzed by using FlowJo 9.9 (Tree Star, Ashland, OR, USA).

### Flux analysis

The oxygen consumption rate and extracellular acidification rate were measured by using a flux analyzer (Seahorse Bioscience XF24 Extracellular Flux Analyzer, Agilent, Santa Clara, CA, USA) and XF Mito Stress Kit (Agilent). 3T3L1 cells were seed at $3.5 × 10^3$ cells per well in 0.1% gelatin-coated Seahorse 24-well plates and differentiated into adipocytes as described above. After differentiation, *B. wexlerae* culture supernatant or uncultured fresh medium were added to the 3T3L1 adipocyte cultures to a final concentration of 10%; cultures were then incubated at 37 °C for 1 h. After incubation, the culture medium was changed to XF Base Medium Minimal DMEM (Agilent) supplemented with 10 mM glucose (Nacalai Tesque), 1 mM pyruvate (Nacalai Tesque), and 2 mM L-glutamine (Nacalai Tesque) for measurement. Compounds injected during the assay and their final concentrations were 1.5 μM oligomycin (inhibitor of ATP synthase), 1 μM FCCP (proton uncoupling agent), and 0.5 μM rotenone + 0.5 μM antimycin A (inhibitors of the mitochondrial respiration complex). XFe Wave software (Agilent) was used to analyze the results.

### Oil red O staining

For measurement of lipid accumulation, 3T3L1 adipocytes were washed with PBS for 1 min, treated with 4% paraformaldehyde in PBS (Nacalai Tesque) at 37 °C for 30 min, washed with 60% (vol/vol) 2-propanol (Nacalai Tesque), and then dried at 37 °C for 15 min. The fixed cells were stained with 0.3% (wt/vol) oil red O (Sigma-Aldrich) in 60% (vol/vol) 2-propanol at 37 °C for 30 min, washed with distilled water three times, and then incubated with 100% 2-propanol at room temperature for 1 min. Absorbance at 490 nm was measured by using an iMark microplate reader (Bio-Rad, Hercules, CA, USA).

### LC−MS/MS

Hydrophilic metabolites were extracted as previously described with minor modification[104,105]. The eAT MAF (100 μl suspended in PBS), serum (50 μl), and bacteria-cultured medium (100 μl) were diluted with water (Wako Pure Chemicals) to a total volume of 200 μl, mixed with 400 μl of methanol (Wako Pure Chemicals) containing methionine sulfone (Wako Pure Chemicals) as an internal standard, and combined with 400 μl of chloroform (Nacalai Tesque). Samples of liver and gastrocnemius muscle were homogenized in methanol (400 μl) containing the internal standard (concentration, 100 mg/ml), and each suspension was mixed with 200 μl of water and 400 μl of chloroform.

Samples were centrifuged at $20,000 × g$ at 4 °C for 15 min, after which 200 μl of supernatant was centrifugally filtered through a 5-kDa cutoff filter (Human Metabolome Technologies, Inc., Tokyo, Japan). The filtrates were lyophilized, resuspended in ultrapure water (Wako Pure Chemicals), and analyzed by LC−MS/MS as previously described[104] by using a Nexera system (Shimadzu, Kyoto, Japan) equipped with two LC-40D pumps, a DGU-405 degasser, an SIL-40C autosampler, a CTO-40C column oven, and a CBM-40 control module, coupled to an LCMS-8050 triple-quadrupole mass spectrometer (Shimadzu). A pentafluorophenylpropyl column (Discovery HS F5, 150 mm × 2.1 mm, 3 μm; Sigma-Aldrich) was used for the separation of metabolites. Instrument control and data analysis were performed by using the software LabSolutions LCMS with LC/MSMS Method Package for Primary Metabolites, ver. 2 (Shimadzu).

### Raman spectroscopic analysis

Raman spectroscopic analysis was performed as described previously with minor modification[106]. Briefly, bacterial cells were cultured anaerobically for 48 h at 37 °C, collected by centrifugation at $10,000 × g$ for 10 min at 4 °C, and suspended in PBS. The cell suspension was analyzed under a confocal Raman microspectroscopic system, which was equipped with a 100×/1.4 N.A. objective lens. The spatial resolution of the system is 0.3 × 0.3 μm laterally and 2.6 μm in depth. The excitation wavelength was 532 nm, and the intensity was 5 mW at the sample

point. The mapping scan measurements were performed in steps of 0.25 μm, with the exposure time of 1 s per point.

The data was preprocessed using IGOR Pro software (Wave-Metrics, Inc., Lake Oswego, OR, USA) for wavenumber calibration and sensitivity correction. Subsequently, a multivariate curve resolution analysis was performed on the spectral data set obtained from all the mapping measurements[107]. In this analysis, the decomposed Raman spectra and their corresponding intensity images were calculated by alternating least-squares optimization, based on the initial value spectra created by concatenating the results of the singular value decomposition (SVD) and the reference spectra of glass and PBS.

### Measurement of amylose and amylopectin
Bacterial cells were cultured anaerobically for 48 h at 37 °C, then collected by centrifugation at 10,000 × g for 10 min at 4 °C, and frozen at −80 °C, and lyophilized (EYELA FDU-2110, Tokyo Rikakikai Co., Ltd, Tokyo, Japan). Amylose and amylopectin contents were measured by using an Amylose/Amylopectin Assay kit (Megazyme, Bray, Ireland) in accordance with the manufacturer's instructions.

### SCFA measurement by HPLC
Murine fecal samples were mixed with 95% ethanol (Nacalai Tesque) to a concentration of 100 mg/ml and homogenized by using two 15-s pulses at 6500 rpm from a tissue homogenizer (Precellys 24, Bertin Instruments, Montigny-le-Bretonneux, France) with zirconia beads (Tomy Digital Biology Co., Ltd, Tokyo, Japan). The homogenate was centrifuged at 1600 × g for 10 min at 4 °C and the supernatant collected. The fecal supernatant or bacterial cultured medium were labeled using an FA Labeling kit (YMC Co., Ltd, Kyoto, Japan) in accordance with the manufacturer's instructions. The labeled samples were analyzed on an HPLC system (Ultimate 3000, Thermo Fisher Scientific, Waltham, Massachusetts, USA) with a 6.0 × 250 mm YMC-Pack FA column (YMC), and the UV spectrum at 400 nm was measured.

### Blautia-specific quantitative PCR analysis
Quantitative PCR analysis was performed by using a LightCycler 480 II (Roche) with Real-time PCR Detection Kit (Blautia: RI-0008) (TechnoSuruga Laboratory Co., Ltd) and TB Green Premix Ex Taq II (Takara Bio) in accordance with the manufacturer's instructions for templates of human and mouse fecal DNA (25 ng).

### Measurement of glucagon-like peptide-1 (GLP-1)
Serum was collected from mice at 10 weeks when they were euthanized for tissue sampling and stored at −80 °C until use. Serum GLP-1 was measured by using LBIS GLP-1 (Active) ELISA Kit (FUJIFILM Wako Shibayagi, Gunma, Japan) in accordance with the manufacturer's instructions.

### Measurement of energy excretion in mouse feces
Fecal samples were collected at 8 weeks, frozen at −80 °C, and lyophilized (EYELA FDU-2110). The lyophilized samples were completely combusted in a Calorimeter C5003 (IKA Japan K.K., Osaka, Japan), and the energy value (J) was calculated according to the amount of heat released during the combustion.

### Measurement of spontaneous activity of mice
Physical spontaneous activity was measured as the number of infrared beams broken in both X and Y directions by using an activity monitoring system (ACTIMO-100, Shinfactory, Fukuoka, Japan). At 8 weeks, mice were placed in the individual cages, and movement was monitored for 24 h.

### Statistics and reproducibility
The output of the QIIME pipeline in Biom table format was imported and analyzed in R (version 3.5.1) (https://www.R-project.org/). The alpha-diversity indices were calculated by using the *estimate_richness* function in the R package 'phyloseq'[108]. The beta-diversity index, calculated according to the Bray–Curtis distance of genus-level data, was generated by using the *vegdist* function in the R package 'vegan'[109]. Principal coordinate analysis (PCoA) was performed by using the *dudi.pco* function in the R package 'ade4'[110], and the PCoA figure was created by using the R package 'ggplot2'[111]. Covariates of gut microbiome beta-diversity variation were identified by calculating the association between continuous or categorical phenotypes and the coordinates of genus-level communities with the *envfit* function in the 'vegan' R package. Multiple-regression analysis by forward selection and single regression analysis was used to identify the gut microbiota associated with BMI (i.e., *lm* function and step function in the R package 'stats'). Multiple logistic regression analysis by forward selection and single logistic regression analysis were used to identify the gut microbiota associated with T2DM (i.e., *glm* function and step function in the R package 'stats'). Heatmaps were created by using the R packages 'corrplot' (https://github.com/taiyun/corrplot) and 'superheat'[112]. The area under the curve was calculated by using Prism 7 (GraphPad Software, La Jolla, California, USA). Statistical significance was evaluated through one-way ANOVA or two-way ANOVA for comparison of multiple groups and the Mann–Whitney $U$-test for two groups by using Prism 7 or R package. A $P$ value less than 0.05 was considered to be significant.

No statistical method was used to predetermine sample size. The experiments were not randomized. The Investigators were not blinded to allocation during experiments and outcome assessment.

### Reporting summary
Further information on research design is available in the Nature Research Reporting Summary linked to this article.

## Data availability
DNA sequencing data generated in this study have been deposited in the DNA Databank of Japan (DDBJ) under the accession numbers DRA010841, DRA012134, and DRA010839. All other data are available in the article and Supplementary Information files. Source data are provided with this paper.

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

## Acknowledgements

We thank Dr. Tokiko Tabata (Kobe University, Japan) for helpful discussion regarding T2DM and its medication; Mr. Motonobu Sato and Dr. Arihiro Kohara (NIBIOHN) for storing samples; staff in Shunan City, Shunan City Shinnanyo Hospital, and Minamiuonuma City for sample collection; and members of our laboratories for helpful discussion. This work was supported by the Ministry of Education, Culture, Sports, Science and Technology of Japan (MEXT)/Japan Society for the Promotion of Science KAKENHI (grant numbers 18K17997 and 22K15004 to K.H.; 18H02674, 20H05697, 20K08534, 20K11560, 18H02150, and 17H04134 to J.K.; 16K00944 and 20H04117 to M.M.; 19K07617 to T.N.; and 21K15267 to J.P.); the Japan Agency for Medical Research and Development (AMED; grant numbers 22ae0121035s0102 to K.H.; 22gm1010006h0004, 22ae0121042h0002, and 22ae0121035s0102 to J.K.; and 17ek0210078h0002 to H.M.); the Ministry of Health and Welfare of Japan and Public /Private R&D Investment Strategic Expansion PrograM: PRISM (grant number 20AC5004 to J.K.); the Ministry of Health, Labour, and Welfare of Japan (grant numbers JP19KA3001 to K.H.; and 201709002B to MM); Cross-ministerial Strategic Innova-tion Promotion Program: SIP (grant number 18087292 to J.K.); the Grant for Joint Research Project of the Institute of Medical Science, the University of Tokyo (to J.K.); the Ono Medical Research Foun-dation (to J.K.); and the Canon Foundation (to J.K.).

## Author contributions

K.H. and J.K. designed the study and wrote the manuscript; K.H., H.M., K.K., H.O., K.T., A. Matsutani, and M.M. collected human samples; K.H., J.P., A. Mohsen, Y.A.C., H.K., Y.N.K., and K.M. performed microbiome analysis; K.H., M.S., Y.O., H.S., M.F., Y.T., S.K., and Y.Y. performed animal and bacterial experiments; N.S., M.A., and H.T. performed Raman spectroscopic analysis; and K.H., M.F., Y.T., T.N., K.S., and A.S. per-formed metabolome analysis.

## Competing interests

The authors of this manuscript have the following potential conflicts of interest: M.S., Y.O., H.S., and Y.Y. are employees of Noster, Inc. (Kyoto, Japan). Other authors declare no competing interests.

## Additional information

[1]Laboratory of Vaccine Materials, Center for Vaccine and Adjuvant Research, National Institutes of Biomedical Innovation, Health and Nutrition (NIBIOHN), 7-6-8 Saito-Asagi, Ibaraki, Osaka 567-0085, Japan. [2]Laboratory of Gut Environmental System, Collaborative Research Center for Health and Medicine, National

Institutes of Biomedical Innovation, Health and Nutrition (NIBIOHN), 7-6-8 Saito-Asagi, Ibaraki, Osaka 567-0085, Japan. [3]Noster Inc., 35-3 Minamibiraki, Kamiueno-cho, Muko, Kyoto 617-0006, Japan. [4]Laboratory of In-silico Drug Design, Artificial Intelligence Center for Health and Biomedical Research, National Institutes of Biomedical Innovation, Health and Nutrition (NIBIOHN), 7-6-8 Saito-Asagi, Ibaraki, Osaka 567-0085, Japan. [5]Department of Physical Activity Research, National Institutes of Biomedical Innovation and Nutrition (NIBIOHN), 1-23-1 Toyama, Shinjuku, Tokyo 162-8636, Japan. [6]Faculty of Sport and Health Science, Ritsumeikan University, 1-1-1 Nojihigashi, Kusatsu, Shiga 525-0085, Japan. [7]Research Organization for Nano and Life Innovation, Waseda University, 513 Waseda-Tsurumaki-cho, Shinjuku, Tokyo 162-0041, Japan. [8]Faculty of Food and Nutritional Sciences, Toyo University, 1-1-1 Izumino, Itakura, Oura, Gunma 374-0193, Japan. [9]Department of Nutrition, Kiryu University, 606-7 Azami, Kasakake-machi, Midori, Gunma 379-2392, Japan. [10]Faculty of Sport Sciences, Waseda University, 2-579-15 Mikajima, Tokorozawa, Saitama 359-1192, Japan. [11]Laboratory of Bioinformatics, Artificial Intelligence Center for Health and Biomedical Research, National Institutes of Biomedical Innovation Health and Nutrition (NIBIOHN), 7-6-8 Saito-Asagi, Ibaraki, Osaka 567-0085, Japan. [12]Graduate School of Pharmaceutical Sciences, Osaka University, 1-6 Yamadaoka, Suita, Osaka 565-0871, Japan. [13]Department of Life Science and Medical Bioscience, Waseda University, 2-2 Wakamatsu-cho, Shinjuku, Tokyo 162-8480, Japan. [14]Computational Bio Big-Data Open Innovation Laboratory, AIST-Waseda University, 3-4-1 Okubo, Shinjuku, Tokyo 169-8555, Japan. [15]Institute for Advanced Research of Biosystem Dynamics, Waseda Research Institute for Science and Engineering, 3-4-1 Okubo, Shinjuku, Tokyo 169-8555, Japan. [16]Shunan City Shinnanyo Hospital, 2-3-15 Miyanomae, Shunan, Yamaguchi 746-0017, Japan. [17]Institute for Protein Research, Osaka University, 3-2 Yamadaoka, Suita, Osaka 565-0871, Japan. [18]International Vaccine Design Center, Institute of Medical Science, University of Tokyo, 4-6-1 Shirokanedai, Minato, Tokyo 108-8639, Japan. [19]Graduate School of Medicine, Osaka University, 2-2 Yamadaoka, Suita, Osaka 565-0871, Japan. [20]Graduate School of Dentistry, Osaka University, 1-8 Yamadaoka, Suita, Osaka 565-0871, Japan. [21]Department of Microbiology and Immunology, Graduate School of Medicine, Kobe University, 7-5-1 Kusunoki-cho, Chuo, Kobe, Hyogo 650-0017, Japan. [22]Graduate School of Science, Osaka University, 1-1 Machikaneyamacho, Toyonaka, Osaka 560-0043, Japan. ✉e-mail: kunisawa@nibiohn.go.jp

