## [Peer Review File · Nature Communications]

REVIEWER COMMENTS

Reviewer #1 (Remarks to the Author):

The manuscript seeks to first correlate and then describe the presence and action of select bacterial species with obesity (and potentially type 2 diabetes) in the gut microbiome. The work starts by analysis of gut microbiome data from a cross-sectional study of 217 Japanese adults. From this assessment, a subset of the statistically implicated organisms are further explored using a series of murine models and then specific species are further explored for their specific attributes. The species *Blautia wexlerae* becomes the primary focus of the paper (always in reference to several other bacteria) to delineate some unique features that can be correlated with attributes that could contribute to obesity and type 2 diabetes.

The experiments are overall quite well reasoned and strong connections are able to be made between the data and the rationale for each subsequent series of experiments presented in the manuscript. The manuscript is well written and logically presented. Data presented in Fig 2I&J, 3D, Fig S4, Fig S6, Fig S9, and Fig S10 show the extremely nice incorporation of nice experiments yielding data that is supportive of the primary data associated with these experiments.

The Raman data presented as part of Figure 5 is nicely incorporated into the manuscript. Using Raman in this manner to highlight distinct features such as protein vs. amyloid carbohydrates takes advantage of the inherent strengths of this technique—to discern Raman light scattering between different types of chemical bonds. This is well differentiated from the LC-MS/MS data that can be used more readily to distinguish different organic acids in the manner used here.

Specific points of criticism:

Some consideration should be given to the fact that the species identified in the cross-sectional study were not those used in the subsequent experiments. Some specific strains may exhibit different metabolic pathways from those elucidated as part of Figures 3 and 4.

While statistically significant, the data presented in Figure 3F and text in lines 261-269 is not very strong that there are differential levels of isocitrate and succinate as compared to lactate and citrate for the *B. wexlerae* supplemented mice.

It would not be possible to actually discern the pathways presented in Fig 4A and Fig S7 from the panels in the manuscript document.

Reviewer #2 (Remarks to the Author):

General comment

This is an interesting piece of work showing that the bacterium *B. wexlerae*, a commensal bacterium that is inversely correlated with obesity and T2D in Japanese adults, can be harnessed to protect against diet-induced metabolic disorders in a mouse model. Indeed, oral administration of *B. wexlerae* to mice appears to promote metabolic and anti-inflammatory effects that alleviated high-fat diet induced obesity/diabetes, apparently by some unique amino-acid metabolism to produce S-adenosylmethionine, acetylcholine, and L-ornithine. Other data suggest that the bacterium also mediated its beneficial effects indirectly by carbohydrate metabolism that resulted in the build-up of amylopectin and production of succinate, lactate, and acetate, in line with changes in gut microbiota. These findings are interesting and the overall approach is commendable. Some novel host-microbial mechanisms are proposed to explain the benefits conferred by this interesting bacterium leading the authors to propose that this work may provide novel therapeutic strategies for mitigating metabolic disorders.

While this reviewer finds the study to be of potential interest to the field, there are, however, several issues and a certain lack of scientific rigor that reduce my overall enthusiasm for the work, as detailed below for individual sections.

Introduction

The introduction is way too long and written in the form of a «mini-review» on obesity. It is not necessary to have a full paragraph to explain the physiology of a dysfunctional WAT or gut microbial dysbiosis and the role of metabolites they produced in the setting of obesity. Please introduce the main topics using key points and leading to the rationale and key hypotheses to be tested in this work.

Additionally, the introduction leaves the reader with the impression that adipose tissue is the only tissue involved in the regulation of glucose homeostasis which is obviously incorrect. This actually announces one important gap in this paper. Indeed, it is not clear why the authors limited their exploration of the mechanism of action of *B. wexlerae* to the study of adipose tissue and not considering other potential mechanisms in liver, skeletal muscle and many other tissues involved in metabolic homeostasis.

Conversely, several topics that are presented in the introduction (eg. bile acids, GLP-1) were neither measured nor considered further in the study, which is somewhat deceiving.

Methodology

Several concerns need to be addressed and/or clarified:

- Cross-sectional human study so data taking only at one time point in participants. This is a limitation that needs to be acknowledged in the discussion.
- Not clear why 25 participants with type 1 diabetes were included in the analysis out of the 70 diabetes participants analysed? It is clearly mentioned in the abstract and the introduction that we are looking at correlations between commensal bacteria and T2D/obesity so the inclusion of several participants with T1D, which is a very different pathology and with distinct etiological mechanisms, is difficult to understand.
- When in the day were the fecal samples collected? were the participants fasted?
- In the methods for animal protocols, it is written that mice were fed a HFHS diet for 2 to 3 months. Why this gap of 4 weeks for HFHS feeding since the metabolic and inflammatory perturbations can change a lot between 2 and 3 months of diet-induced obesity.
- Did mice receive the bacterial solution for the same amount of time (2-3 months)? In figure 2 it is written that mice were maintained for 8 weeks on standard chow (control diet, CD) or HFD without or with oral administration of *B. wexlerae*. This needs to be clarified.
- What is the proportion of macronutrients in the HFHS diet?
- What is the amount of *B. wexlerae* given to the mice (e.g. 10⁹ CFU)? It is only mentioned 0.5ml of bacterial solution. And what is the rationale for only 3 times per week?
- Were the control HFD-fed mice gavaged with a vehicle? This is important since stress related to daily gavage of mice receiving the bacteria alone could decreased weight gain per se.
- In figure 2, the statistical analysis was evaluated using One-way ANOVA. However, the authors have performed repeated measures in figure 2A, G, H. The appropriate statistical tests should be used.
- Why the used of IPGTT and not OGTT if the overall goal was to study the influence of the gut microbiota in this study?
- Fasting mice for 16 hours is very long and even considered as starvation. Please clarify why you fasted so long and mention this limitation in the data interpretation in the discussion since the findings may also reflect metabolic changes that occur during starvation.
- Many key details are missing in the methods which are important for others to be able to reproduce such findings: 1) Were the mice group-housed? 2) Did they had ad libitum access to food and water? 3) How many mice per group? 4) How were the mice euthanized? 5) How were the tissue collected?

- In the cell culture experiments, how many biological and technical replicates were performed?

Results

Some data were clearly overinterpreted:

- For example: line 197-198 – “These results indicate that *B. wexlerae* plays a causative role in the prevention of obesity.” It is a very strong statement considering that the bacterium prevented weight gain in a group of only 5 mice, only performed once. This finding should be reproduced at least in another group of 5 mice to be more convincing. Again were control mice gavaged with vehicle to control for the stress-induced weight loss related to the gavage procedure?

- The authors should be more nuanced in many other statements. e.g. Line 413, I think it would be more appropriate to say Our human cohort study showed that intestinal *B. wexlerae* was associated to a lower risk of obesity and diabetes in Japanese adults. Rather than Our human cohort study showed that intestinal *B. wexlerae* reduces the risk of obesity and diabetes in Japanese adults.

- The diet-induced mouse model qualifies as an insulin-resistant and pre-diabetic model and not T2D as mentioned by the authors.

- Not clear why the number of mice changes so much depending on the analysis? (Figure 2 legend, 5 mice for weight gain, 10 for IPGTT, but when counting the dots, range from 7 to 10. Please clarify why some animals seem to be lacking (or not used) in some analyses.

- immunohistochemistry pictures in panel J of figure 2 appear blurry and it is not possible to discern cell shapes. BODIPY staining also seems to be unspecific.

- For qPCR, relative mRNA to Actinb is uncommon, please refer to the MIQE guidelines.

- It would have been relevant to assess the presence of *B. wexlerae* in the gut microbiota of treated-mice compare to HFHS-fed mice. Moreover, a major concern is that mice were treated with *Blautia*, but when looking at the Lefse analysis, this bacterium is not overrepresented in the treated group compare to the control group? In fact Figure 5F shows that the microbiota of these mice is overrepresented by an increase in the relative abundance of *Akkermansia*... How do you interpret this surprising finding given the major role of *Akkermansia* in metabolic benefits (e.g. Cani and many other groups)?

- It is critical to perform validation by quantification of *B. wexlerae* (e.g. not only the relative abundance) by qPCR in both human and mouse samples?

- Figure 5E: What is the p-value of the statistical difference between HFD and HFD+bw?

- As the chow diet and the HFD does not contain the same amount of kcal/g, please provide energy consumption in kcal (Figure S4).

- What was the rationale to test *Bacteroides vulgatus*, *Prevotella copri*,

Faecalibacterium prausnitzii in figures 4 and 5? Why not go back to the original bacteria that were changed in the human cohort?

- The investigation of the mitochondrial activity in adipose tissue is interesting. However, why authors did not explore this mechanism in more typical oxidative tissues such as muscle and liver, which are playing an even greater role in energy metabolism?

- Other mechanisms that could explain changes in body weight and fat mass are changes in energy excretion and/or expenditure. Do animals excrete more energy in feces when treated with B. wexlerae? Are animals spending more energy when placed in metabolic cages to assess indirect calorimetry?

- This reviewer certainly appreciated the analysis of the composition of the B. wexlerae supernatant in order to better understand its effects on metabolism. However, I would like to know which and how much of these metabolites actually reach the circulation of mice treated with B. wexlerae?

- Also, would it be possible to determine whether T2D subjects from the initial cohort have lower levels of these metabolites compared to healthy subjects?

Minor concerns:

Line 96-97: In several studies using human and animal models, gut microbial transplantation led to the transfer of obesity and T2DM phenotypes (12-14). The evidence of a transmission of obesity by the gut microbiota transfer from one human to another has not been convincingly demonstrated to the best of my knowledge. Authors should be more cautious in their assertion, since only transplantation from a human donor to a recipient mouse has been shown to convincingly and reproductively transfer microbiota-related body weight changes. The authors should also take a particular attention to the references they cited since ref 12-14 are reviews and not original papers. It would be more appropriate to support their affirmation by original publications.

I believe that the 2D and 2E panels are redundant since we can see these values in figure 2G and 2H in addition to the HOMA-IR calculation in 2F.

Reviewer #3 (Remarks to the Author):

This study starts with a cross-sectional 16s analysis of fecal samples from 217 Japanese adults. The goal is to identify a possible association between gut microbiota and obesity or type 2 diabetes. The authors use multiple and single logistic regression and identify a number of genera to be associated

with these diseases. Because the odds ratio of *Blautia* abundance was inversely correlated with obesity and type 2 diabetes mellitus, and further analysis identified similar associations with *Blautia wexlerae*, the authors focus the remainder of the study on *Blautia wexlerae*.

They next describe that repeated oral gavages (3-times a week for 2-3 months) of *Blautia wexlerae* to mice during a course of high fat diet resulted in reduced weight gain compared to non-gavage mice. The authors also observe metabolic changes (insulin levels following intraperitoneal glucose injection) and anti-inflammatory effects (reduced cytokine mRNA levels and macrophage counts in epididymal adipose tissue) and propose that these effects contribute to decreased high-fat diet induced obesity and diabetes.

The authors then pursue *in vitro* studies to identify the underlying mechanisms. Specifically, they culture a pre-adipocyte cell line with supernatants from *Blautia wexlerae* cultures and assess the expression of genes that are associated with differentiation to adipocytes. They also assess mitochondrial mass and function. To confirm the conclusion that *Blautia wexlerae* stimulates mitochondrial metabolism, the authors then study the transcriptome of mature adipocytes in the epididymal adipose tissue of high-fat diet fed mice and control mice. Finally, they assess the metabolic profile of *Blautia wexlerae* based on genetic information (KEGG orthologues) and LC-MS/MS analysis of supernatant from *Blautia wexlerae* cultures.

In sum, they attribute the protective effect of *Blautia wexlerae* against obesity and type 2 diabetes mellitus to (a) *Blautia wexlerae* producing S-adenosylmethionine, acetylcholine and L-ornithine (these metabolites were subsequently added to the pre-adipocyte cell line which was found to reduce s100a8 expression (a chemokine recruiting macrophages) and lipid accumulation) and (b) altering the carbohydrate metabolism and the bacterial profile and short-chain fatty acid content of the gut microbiota. The authors suggest that this information may be used for prevention of obesity and diabetes.

Comments to the authors:

Patient data:

1. Please include a validation cohort.
2. How were fecal samples collected and preserved. Were they frozen at the patient's location and shipped or collected in a hospital and/or outpatient setting? If the latter, why did the non-diabetic patients visit the clinic? Please stratify for other diseases and medication.
3. Table S1 does not stratify patients into obese / non-obese subgroups, only for presence or absence of type 2 diabetes. There is a wide range in body mass index in each group. Please stratify

between overweight and obese patients. It appears that most patients are overweight but not obese.

In vivo studies (mouse model):

4. As stated in the methods section, the mice received 3 oral gavages with *Blautia wexlerae* per week during the entire 2-3 months of high fat diet. I did not see any mock gavages in the control group and think that the gavages themselves may have caused changes in weight gain and composition of the gut microbiome (dysbiosis). This is a major concern.

5. Does supplementation of *Blautia wexlerae* prior to high fat diet reduce weight gain?

6. What is the composition of the high fat diet (% calories from fat, fiber content etc)?

7. Figure 2I and 2J: Please show data from more than one mouse per group.

8. The sex of the mice is not specified. Please show data separately for male and female mice.

9. All figure legends need to include statements on number of repeat experiments, number of mice per group

In vitro studies:

10. The claim that *Blautia wexlerae* metabolites affect adipocyte differentiation is based on exposing the 3T3L1 pre-adipocyte cell line to supernatant from *Blautia wexlerae* cultures. I do think that this approach is too limited. In vivo, there is no direct contact between *Blautia wexlerae*, a gut commensal, and adipocytes. Any change in metabolites that is introduced by *Blautia wexlerae* in vivo is much more complex, taking into account changes in the gut microbiome, its metabolites, adipokines, gastrointestinal hormones and so on.

11. The LC-MS/MS experiments were performed on epididymal MAF of mice suspended in phosphate-buffered saline or bacterial cultured medium. It is also not clear to me that phosphate-

buffered is the correct control here – is there any reason not to use culture medium without bacteria or with other bacteria? However, my general concern is that in vivo there is presumably no direct contact between *Blautia wexlerae*, a gut commensal, and fat tissue.

12. The claim that *Blautia wexlerae* metabolites confer anti-inflammatory effects to adipose tissue should be confirmed by ex vivo analysis of cytokine expression and analysis of immune cell composition and function. This is important as the authors specify reduced s100a8 mRNA expression (a chemokine recruiting macrophages) and reduced mRNA levels of TNF α in adipose tissue.

National Institutes of Biomedical
Innovation, Health and Nutrition

Laboratory of Vaccine Materials, Center for Vaccine and Adjuvant Research,
Laboratory of Gut Environmental System, Collaborative Research Center for
Health and Medicine

7-6-8 Asagi Saito, Ibaraki-shi, Osaka 567-0085, Japan
Tel: 81-72-641-9871, Fax: 81-72-641-9872

27 May 2022

Dear Reviewers,

Thank you for reviewing our manuscript, entitled “**Unique metabolic profiles of *Blautia wexlerae* achieve beneficial effects for the control of obesity and type 2 diabetes**” (NCOMMS-21-43496). We appreciate the constructive and positive comments from the reviewers and have revised the manuscript accordingly, the changes to which are in red font. Our point-by-point responses to the reviewers’ comments follow.

I respectfully request your reconsideration regarding the publication of our revised manuscript in *Nature Communications*.

Sincerely yours,

Jun Kunisawa, Ph.D.
Director

Point-by-point response to the reviewers' comments

Reviewer #1 (Remarks to the Author):

*The manuscript seeks to first correlate and then describe the presence and action of select bacterial species with obesity (and potentially type 2 diabetes) in the gut microbiome. The work starts by analysis of gut microbiome data from a cross-sectional study of 217 Japanese adults. From this assessment, a subset of the statistically implicated organisms are further explored using a series of murine models and then specific species are further explored for their specific attributes. The species *Blautia wexlerae* becomes the primary focus of the paper (always in reference to several other bacteria) to delineate some unique features that can be correlated with attributes that could contribute to obesity and type 2 diabetes.*

The experiments are overall quite well reasoned and strong connections are able to be made between the data and the rationale for each subsequent series of experiments presented in the manuscript. The manuscript is well written and logically presented. Data presented in Fig 2I&J, 3D, Fig S4, Fig S6, Fig S9, and Fig S10 show the extremely nice incorporation of nice experiments yielding data that is supportive of the primary data associated with these experiments.

The Raman data presented as part of Figure 5 is nicely incorporated into the manuscript. Using Raman in this manner to highlight distinct features such as protein vs. amyloid carbohydrates takes advantage of the inherent strengths of this technique—to discern Raman light scattering between different types of chemical bonds. This is well differentiated from the LC-MS/MS data that can be used more readily to distinguish different organic acids in the manner used here.

Thank you very much for your efforts in reviewing our manuscript, and we appreciate your positive and constructive comments.

Specific points of criticism:

Some consideration should be given to the fact that the species identified in the cross-sectional study were not those used in the subsequent experiments. Some specific strains may exhibit different metabolic pathways from those elucidated as part of Figures 3 and 4.

We agree with the reviewer's important point. Even when they are of the same species, different strains may differ genetically and functionally. To address this issue, we are now using single-cell analysis for *Blautia wexlerae* in human fecal samples. Our preliminary data suggest that there is a degree of genomic diversity of *B.wexlerae* strains, which can be divided into 3 clusters and may be related to their metabolic

function. However, these data are too premature to be published in this paper. Therefore, we hope that you understand our preference to address the functional differences between bacterial strains in future research. Thank you again for your suggestion, and we have discussed this important point in lines 566–569 of the revised manuscript.

While statistically significant, the data presented in Figure 3F and text in lines 261-269 is not very strong that there are differential levels of isocitrate and succinate as compared to lactate and citrate for the B. wexlerae supplemented mice.

As you point out, when considering the lactate and citrate levels in *B. wexlerae* supplemented mice, we should understand that the administration of *B. wexlerae* only moderately stimulates energy metabolism. We have revised this claim so that we do not overstate these data (lines 253–260 of the revised manuscript).

It would not be possible to actually discern the pathways presented in Fig 4A and Fig S7 from the panels in the manuscript document.

We apologize for this difficulty. We have enlarged the panels as much as possible and have modified these pictures, which are shown as Fig. S17–S21 of the revised manuscript.

Reviewer #2 (Remarks to the Author):

General comment

*This is an interesting piece of work showing that the bacterium *B. wexlerae*, a commensal bacterium that is inversely correlated with obesity and T2D in Japanese adults, can be harnessed to protect against diet-induced metabolic disorders in a mouse model. Indeed, oral administration of *B. wexlerae* to mice appears to promote metabolic and anti-inflammatory effects that alleviated high-fat diet induced obesity/diabetes, apparently by some unique amino-acid metabolism to produce S-adenosylmethionine, acetylcholine, and L-ornithine. Other data suggest that the bacterium also mediated its beneficial effects indirectly by carbohydrate metabolism that resulted in the build-up of amylopectin and production of succinate, lactate, and acetate, in line with changes in gut microbiota. These findings are interesting and the overall approach is commendable. Some novel host-microbial mechanisms are proposed to explain the benefits conferred by this interesting bacterium leading the authors to*

propose that this work may provide novel therapeutic strategies for mitigating metabolic disorders.

While this reviewer finds the study to be of potential interest to the field, there are, however, several issues and a certain lack of scientific rigor that reduce my overall enthusiasm for the work, as detailed below for individual sections.

Thank you very much for your efforts in reviewing our manuscript, and we appreciate your important and constructive comments.

Introduction

The introduction is way too long and written in the form of a «mini-review» on obesity. It is not necessary to have a full paragraph to explain the physiology of a dysfunctional WAT or gut microbial dysbiosis and the role of metabolites they produced in the setting of obesity. Please introduce the main topics using key points and leading to the rationale and key hypotheses to be tested in this work.

We have modified the Introduction accordingly.

*Additionally, the introduction leaves the reader with the impression that adipose tissue is the only tissue involved in the regulation of glucose homeostasis which is obviously incorrect. This actually announces one important gap in this paper. Indeed, it is not clear why the authors limited their exploration of the mechanism of action of *B. wexlerae* to the study of adipose tissue and not considering other potential*

mechanisms in liver, skeletal muscle and many other tissues involved in metabolic homeostasis.

We apologize for the insufficient description in the Introduction. Accordingly, we have deleted redundant and biased representations regarding the role of adipose tissue in the regulation of glucose homeostasis. Furthermore, we have performed additional analyses of liver and muscle samples; we now explain these data in the Results section (lines 261–270 of the revised manuscript). Thank you again for your important comment.

Conversely, several topics that are presented in the introduction (eg. Bile acids, GLP-1) were neither measured nor considered further in the study, which is somewhat deceiving.

As the reviewer suggested, we have deleted these comments from the Introduction.

Methodology

Several concerns need to be addressed and/or clarified:

- Cross-sectional human study so data taking only at one time point in participants. This is a limitation that needs to be acknowledged in the discussion.

We agree and now describe the limitations of our human study in the Discussion (lines 553–557 of the revised manuscript).

- Not clear why 25 participants with type 1 diabetes were included in the analysis out of the 70 diabetes participants analysed? It is clearly mentioned in the abstract and the introduction that we are looking at correlations between commensal bacteria and T2D/obesity so the inclusion of several participants with T1D, which is a very different pathology and with distinct etiological mechanisms, is difficult to understand.

Both T1D and T2D are included in this study because of the recruitment of diabetic patients at the hospital. In the analysis of the relationship between BMI (obesity) and intestinal bacteria (Fig. 1A), our exclusion of T1D was somewhat arbitrary. However, because T1D and T2D differ in pathogenesis, we chose to focus on T2D from the viewpoint of diabetes, exclude T1D, and analyze the relationship between T2D and intestinal bacteria (Fig. 1B). We have added this explanation to the legend for Fig. 1 (lines 963–971 of the revised manuscript) and Table S1, S3, S4.

- When in the day were the fecal samples collected? were the participants fasted?

Participants were under no restrictions (e.g., fasting), collected their stool sample at home, and submitted it to the hospital or health examination site within 5 days after collection. We now include this information in lines 599–603 of the revised manuscript.

- In the methods for animal protocols, it is written that mice were fed a HFHS diet for 2 to 3 months. Why this gap of 4 weeks for HFHS feeding since the metabolic and inflammatory perturbations can change a lot between 2 and 3 months of diet-induced obesity.

We apologize for providing insufficient information regarding the mouse experiments. We have described the timeline for each experiment in lines 660–673 of the revised manuscript and showed the experimental schedule in Fig. S6A.

- Did mice receive the bacterial solution for the same amount of time (2-3 months)? In figure 2 it is written that mice were maintained for 8 weeks on standard chow (control diet, CD) or HFD without or with oral administration of B. wexlerae. This needs to be clarified.

Mice in treatment groups received the bacterial solution until they were used in experiments. We now provide the experimental schedule in Fig. S6 to clarify the protocol.

- What is the proportion of macronutrients in the HFHS diet?

We have added this information as Fig. S7A in the revised manuscript.

- What is the amount of B. wexlerae given to the mice (e.g. 10⁹ CFU)? It is only mentioned 0.5ml of bacterial solution. And what is the rationale for only 3 times per week?

We include the dose of *B. wexlerae* (5×10^8 CFU) in line 666 of the revised manuscript. We used the same dosing schedule as in our previous study (Nagatake T. et al., Mucosal Immunol. 15, 289–300, 2022), which used the same model, but the optimal frequency and dose of administration for *B. wexlerae* should be evaluated in future study.

- Were the control HFD-fed mice gavaged with a vehicle? This is important since stress related to daily gavage of mice receiving the bacteria alone could decreased

weight gain per se.

Yes, the untreated HFD-fed mice received oral gavage of fresh medium as a vehicle-only control. We apologize for omitting this information and now include it in lines 666–667 of the revised manuscript.

- In figure 2, the statistical analysis was evaluated using One-way ANOVA. However, the authors have performed repeated measures in figure 2A, G, H. The appropriate statistical tests should be used.

To address this point, we re-analyzed by using two-way ANOVA (Fig. 2A), calculated the area under the curve (AUC), and then repeated the statistical analysis (Fig. 2E, 2F).

- Why the used of IPGTT and not OGTT if the overall goal was to study the influence of the gut microbiota in this study?

OGTT also takes into account intestinal absorption, but IPGTT reflects changes in metabolic function. We selected IPGTT for this study because decreases in obesity suggest changes in metabolic function.

- Fasting mice for 16 hours is very long and even considered as starvation. Please clarify why you fasted so long and mention this limitation in the data interpretation in the discussion since the findings may also reflect metabolic changes that occur during starvation.

We fasted the mice according to the methodology in previous reports (Matsuzaka, T., et al., Nat. Med. 13, 1193–1202, 2007; Nagatake T. et al., Mucosal Immunol. 15, 289–300, 2022). But the reviewer's concern is valid, so we now address this point in the Discussion (lines 560–562 of the revised manuscript).

- Many key details are missing in the methods which are important for others to be able to reproduce such findings: 1) Were the mice group-housed? 2) Did they had ad libitum access to food and water? 3) How many mice per group? 4) How were the mice euthanized? 5) How were the tissue collected?

We apologize for the insufficient information regarding the mouse experiments. We now address these points in the revised manuscript as follows:

1) Each experimental group comprised 5 mice, which were maintained in a single cage (line 661).

2) Mice had unrestricted access to food and water (lines 665–666).

- 3) Each cage held 5 mice (line 661).
- 4) Mice were euthanized by using cervical dislocation (lines 672–673).
- 5) After mice were euthanized, their tissues were collected by using surgical scissors (lines 672–673).

- In the cell culture experiments, how many biological and technical replicates were performed?

We performed at least two independent experiments and include this information in the figure legends.

Results

Some data were clearly overinterpreted:

- For example: line 197-198 – “These results indicate that B. wexlerae plays a causative role in the prevention of obesity.” It is a very strong statement considering that the bacterium prevented weight gain in a group of only 5 mice, only performed once. This finding should be reproduced at least in another group of 5 mice to be more convincing. Again were control mice gavaged with vehicle to control for the stress-induced weight loss related to the gavage procedure?

All experiments, including the murine experiments in Fig. 2A–2C, were performed independently at least twice, and the reproducibility of the results was confirmed; this information is now in the figure legends. As mentioned earlier, the untreated HFD-fed mice received oral gavage with fresh medium as a vehicle-only control. To avoid overinterpreting our data, we have modified the text in lines 182–183 of the revised manuscript.

- The authors should be more nuanced in many other statements. e.g. Line 413, I think it would be more appropriate to say Our human cohort study showed that intestinal B. wexlerae was associated to a lower risk of obesity and diabetes in Japanese adults. Rather than Our human cohort study showed that intestinal B. wexlerae reduces the risk of obesity and diabetes in Japanese adults.

Thank you for suggesting a more accurate description. We have modified the text accordingly (lines 452, 453, and 460 of the revised manuscript).

- The diet-induced mouse model qualifies as an insulin-resistant and pre-diabetic model and not T2D as mentioned by the authors.

We agree with this comment and have clearly defined the model as such in the

Methods (lines 664–665) and address this point in the Discussion (lines 557–560).

- Not clear why the number of mice changes so much depending on the analysis? (Figure 2 legend, 5 mice for weight gain, 10 for IPGTT, but when counting the dots, range from 7 to 10. Please clarify why some animals seem to be lacking (or not used) in some analyses.

For all experiments, groups of 5 mice were maintained and the experiment repeated, but sample numbers are reduced in some cases because unsuitable specimens, such as hemolyzed blood, were excluded from analysis. We include this explanation in line 687 of the revised manuscript.

- immunohistochemistry pictures in panel J of figure 2 appear blurry and it is not possible to discern cell shapes. BODIPY staining also seems to be unspecific.

We repeated the experiment and now provide a new picture in Fig. 2G of the revised manuscript. The fluorescence area in each image was calculated by using the software provided with the BZ-X800 analyzer (version 1.1.2.4; Fig. S9). In addition, another reviewer commented on the evaluation of inflammation in adipose tissue, so we have added flow cytometric data (Fig. 2H, S10) to support these findings. Collectively, our results suggest that the administration of *B. wexlerae* reduced inflammation in adipose tissue.

- For qPCR, relative mRNA to Actinb is uncommon, please refer to the MIQE guidelines.

According to the MIQE guidelines, we have modified the text to ‘Relative expression normalized to *Actb*’ in the Figures and Methods (lines 759–760 of the revised manuscript).

- It would have been relevant to assess the presence of B. wexlerae in the gut microbiota of treated-mice compare to HFHS-fed mice. Moreover, a major concern is that mice were treated with Blautia, but when looking at the Lefse analysis, this bacterium is not overrepresented in the treated group compare to the control group? In fact Figure 5F shows that the microbiota of these mice is overrepresented by an increase in the relative abundance of Akkermansia... How do you interpret this surprising finding given the major role of Akkermansia in metabolic benefits (e.g. Cani and many other groups)?

Thank you for the important suggestion. Accordingly, we now show *Blautia*

abundance in mice as Fig. S27A. The proportion of *Blautia* was decreased in HFD-fed mice compared with CD-fed mice; this is consistent with human data, which show that the proportion of *Blautia* is low during obesity. The administration of *B. wexlerae* did not affect the abundance of *Blautia*. Although this outcome seems to be contradictory to the finding that the proportion of *Blautia* in fecal samples did not increase when *B. wexlerae* was administered, the result is reasonable because the dose of *B. wexlerae* provided is very small relative to that of bacteria present in the intestine. Alternatively, *B. wexlerae* may be unable to colonize and grow in the intestine of mice because the niche already is occupied by intestinal commensal bacteria. We discuss these points in lines 403–409 of the revised manuscript.

Regarding *Akkermansia*, the increase in the relative abundance of *Akkermansia* is very interesting and may be related to metabolic changes in mice. We have added information regarding the role of *Akkermansia* in metabolism (lines 423–428 of the revised manuscript) together with a possible metabolite-driven mechanism, such as lactate- and acetate-mediated interaction between *Blautia* and *Akkermansia*. Thank you again for your important insight.

- It is critical to perform validation by quantification of B. wexlerae (e.g. not only the relative abundance) by qPCR in both human and mouse samples?

We quantified *Blautia* through qPCR analysis. In both human and mouse samples, the data from qPCR analysis paralleled the relative abundance of *Blautia* as calculated through 16S rRNA gene amplicon sequencing analysis. These results are shown in Fig. S2A–S2C (human) and Fig. S27B (mouse) and described in lines 143–146 and 406 of the revised manuscript.

- Figure 5E: What is the p-value of the statistical difference between HFD and HFD+bw?

The *p* value was <0.01 for the comparison between HFD and HFD+Bw, which was calculated by using the envfit function in the ‘vegan’ program in the R package. We now include this information in the Results (line 402) and Methods (lines 907–910) sections of the revised manuscript.

- As the chow diet and the HFD does not contain the same amount of kcal/g, please provide energy consumption in kcal (Figure S4).

We now showed the energy consumption (in kcal) in Fig. S7C and described these data in lines 178–180 of the revised manuscript.

- What was the rationale to test Bacteroides vulgatus, Prevotella copri, Faecalibacterium prausnitzii in figures 4 and 5? Why not go back to the original bacteria that were changed in the human cohort?

Thank you for your suggestion. As described in lines 288–289, we selected *B. vulgatus*, *P. copri*, and *F. prausnitzii* because they were major intestinal genera in the healthy Japanese adults in our previous report (Park J. et al., BMC Microbiol., 2021). However, as the reviewer points out, we agree that it is important to go back to the original human data. When we rechecked these data, these 3 bacteria are the major intestinal bacteria in the current study participants as well. We include the data in Figure S16 and described these findings in lines 288–289 of the revised manuscript.

- The investigation of the mitochondrial activity in adipose tissue is interesting. However, why authors did not explore this mechanism in more typical oxidative tissues such as muscle and liver, which are playing an even greater role in energy metabolism?

Given the decreased adipose tissue weight and suppression of inflammation, we focused on the metabolic function of adipose tissue. We agree that it is important to consider other potential mechanisms in muscle and liver. Indeed, although muscle showed very little metabolic change, the same metabolites showed marked changes in both adipose tissue and liver. In particular, administration of *B. wexlerae* caused succinate accumulation, which is a metabolic signature of thermogenesis, in HFD-fed mice. We now show these data in Fig. S13 and describe them in lines 261–270 of the revised manuscript. Thank you again for your helpful comments.

- Other mechanisms that could explain changes in body weight and fat mass are changes in energy excretion and/or expenditure. Do animals excrete more energy in feces when treated with B. wexlerae? Are animals spending more energy when placed in metabolic cages to assess indirect calorimetry?

To address the possibility regarding energy excretion, because metabolic cages are unavailable in our facility, we instead examined the energy excreted in feces and the spontaneous activity of mice. Administration of *B. wexlerae* had little effect in HFD-fed mice, as shown in Fig. S15A and S15B and described in lines 274–279 of the revised manuscript. In addition, we discuss our technical difficulties in lines 562–566 of the revised manuscript.

- This reviewer certainly appreciated the analysis of the composition of the B. wexlerae supernatant in order to better understand its effects on metabolism. However, I would like to know which and how much of these metabolites actually reach the circulation of mice treated with B. wexlerae?

Accordingly, we attempted to measure serum levels of these metabolites. S-adenosylmethionine and acetylcholine are known to be rapidly eliminated from blood through metabolic degradation, excretion in the urine, and transfer to organs. Indeed, we were unable to detect adenosylmethionine and acetylcholine in serum samples, even when mice were treated with *B. wexlerae*. In contrast, although administration of *B. wexlerae* increased fecal levels of L-ornithine in HFD-fed mice, serum concentrations of L-ornithine were similar among CD-fed mice, untreated HFD-fed mice, and *B. wexlerae*-treated HFD-fed mice. Orally supplemented L-ornithine is converted to several metabolites through host amino acid and urea metabolism, collectively suggesting that exogenous L-ornithine is utilized for host metabolism, rather than circulating in its native form. We show these data in Fig. S23 and described them in lines 327–339 of the revised manuscript.

- Also, would it be possible to determine whether T2D subjects from the initial cohort have lower levels of these metabolites compared to healthy subjects?

As for mouse samples, we were unable to detect any S-adenosylmethionine or acetylcholine in human serum. In contrast, although L-ornithine has been detected in human sera and feces, no relationship between L-ornithine amount and *Blautia* abundance has been identified, probably because several factors including dietary habits, age, and urea cycle influence L-ornithine levels. We provide this explanation in lines 339–345 of the revised manuscript.

However, L-ornithine was increased in the sera of T2D patients compared with healthy subjects (unpublished data). This result was unexpected; age appears to be a confounding factor in T2D, but this association is not yet understood well. Therefore, we intend to address L-ornithine homeostasis in T2D in future research and choose not to report these data in the current study. Thank you again for your suggestion, which led us to this interesting finding.

Minor concerns:

Line 96-97: In several studies using human and animal models, gut microbial

transplantation led to the transfer of obesity and T2DM phenotypes (12-14). The evidence of a transmission of obesity by the gut microbiota transfer from one human to another has not been convincingly demonstrated to the best of my knowledge. Authors should be more cautious in their assertion, since only transplantation from a human donor to a recipient mouse has been shown to convincingly and reproductively transfer microbiota-related body weight changes. The authors should also take a particular attention to the references they cited since ref 12-14 are reviews and not original papers. I would be more appropriate to support their affirmation by original publications.

We have modified the Introduction accordingly.

I believe that the 2D and 2E panels are redundant since we can see these values in figure 2G and 2H in addition to the HOMA-IR calculation in 2F.

We have moved these data to Fig. S8.

Reviewer #3 (Remarks to the Author):

This study starts with a cross-sectional 16s analysis of fecal samples from 217 Japanese adults. The goal is to identify a possible association between gut microbiota and obesity or type 2 diabetes. The authors use multiple and single logistic regression and identify a number of genera to be associated with these diseases. Because the odds ratio of Blautia abundance was inversely correlated with obesity and type 2 diabetes mellitus, and further analysis identified similar associations with Blautia wexlera, the authors focus the remainder of the study on Blautia wexlera.

They next describe that repeated oral gavages (3-times a week for 2-3 months) of Blautia wexlera to mice during a course of high fat diet resulted in reduced weight gain compared to non-gavage mice. The authors also observe metabolic changes (insulin levels following intraperitoneal glucose injection) and anti-inflammatory effects (reduced cytokine mRNA levels and macrophage counts in epididymal adipose tissue) and propose that these effect contribute to decreased high-fat diet induced obesity and diabetes.

The authors then pursue in vitro studies to identify the underlying mechanisms. Specifically, they culture a pre-adipocyte cell line with supernatants from Blautia wexlerae cultures and assess the expression of genes of genes that are associated with differentiation to adipocytes. They also assess mitochondrial ass and function. To confirm the conclusion that Blautia wexlerae stimulates mitochondrial metabolism, the authors then study the transcriptome of mature adipocytes in the epididymal adipose tissue of high-fat diet fed mice and control mice. Finally, they assess the metabolic profile of Blautia wexlerae based on genetic information (KEGG orthologues) and LC-MS/MS analysis of supernatant from Blautia wexlerae cultures. In sum, they contribute the protective effect of Blautia wexlerae against obesity and type 2 diabetes mellitus was attributed to (a) Blautia wexlerae producing S-adenylmethionine, acetylcholine and L-ornithine (these metabolites were subsequently added to the pre-adipocyte cell line which was found to reduce s100a8 expression (a chemokine recruiting macrophages) and lipid accumulation) and (b) altering the carbohydrate metabolism and the bacterial profile and short-chain fatty acid content of the gut microbiota. The authors suggest that this information may be used for prevention of obesity and diabetes.

Thank you very much for your efforts in reviewing our manuscript, and we appreciate your positive and constructive comments.

Comments to the authors:

Patient data:

1. Please include a validation cohort.

Because of the COVID-19 pandemic, we were unable to establish a hospital-based cohort of diabetic patients. Instead, we used a validation cohort comprising subjects from the general Japanese population to confirm the reproducibility of findings regarding obesity and *Blautia*. Indeed, in the validation cohort, the *Blautia* abundance was lower in obese subjects (n = 50) than normal subjects (n = 132). We address these points in lines 157–161 of the revised manuscript, show the results in Fig. S4A–S4C, and provide participant information in Table S8.

2. How were fecal samples collected and preserved. Were they frozen at the patient's location and shipped or collected in a hospital and/or outpatient setting? If the latter, why did the non-diabetic patients visit the clinic? Please stratify for others disease and medication.

We apologize for providing insufficient information. DNA was isolated from fecal samples stored in guanidine thiocyanate solution (TechnoSuruga Laboratory Co., Ltd., Shizuoka, Japan) as previously described (Hosomi K., et al., Sci. Rep. 7, 4339, 2017); this protocol allows the stool sample to be preserved at room temperature. The participants collected their stool sample at home and submitted it to the hospital (diabetic patients) or health examination site (control non-diabetic subjects). We provide this information lines 599–603 of the revised manuscript. Therefore, the control non-diabetic subjects did not visit the hospital for medical treatment, so they are basically healthy and have no disease other than hypertension and dyslipidemia, as shown in Table S1.

3. Table S1 does not stratify patients into obese / non-obese subgroups, only for presence or absence of type 2 diabetes. There is a wide range in body mass index in each group. Please stratify between overweight and obese patients. It appears that most patients are overweight but not obese.

Accordingly, we now have stratified between overweight and obese participants (Table S2). As the reviewer points out, the number of obese participants in this study is low, reflecting the characteristics of the overall Japanese population.

In vivo studies (mouse model):

4. As stated in the methods section, the mice received 3 oral gavages with *Blautia wexlerae* per week during the entire 2-3 months of high fat diet. I did not see any

mock gavages in the control group and think that the gavages themselves may have caused changes in weight gain and composition of the gut microbiome (dysbiosis). This is a major concern.

The untreated HFD-fed mice were received oral gavage of fresh medium as a vehicle-only control (lines 665–667 of the revised manuscript).

5. Does supplementation of Blautia wexlerae prior to high fat diet reduce weight gain?

Supplementation of *B. wexlerae* for 2 weeks prior switching to the high-fat diet did not affect the body weight of mice. We consider that *B. wexlerae* supplementation suppresses excessive weight gain but does not cause pathologic weight loss or growth inhibition. We show these data in Fig. S6B and described them in lines 178–179 of the revised manuscript.

6. What is the composition of the high fat diet (% calories from fat, fiber content etc)?

We apologize for omitting this important information; we now show it in Fig. S7A.

7. Figure 2I and 2J: Please show data from more than one mouse per group.

We have added data from additional mice (Fig. 2G). In addition, to confirm the reproducibility of the histological analysis, we calculated the fluorescence area in each image by using the software provided with the BZ-X800 analyzer (version 1.1.2.4; Fig. S9).

8. The sex of the mice is not specified. Please show data separately for male and female mice.

All mice were male (line 660 of the revised manuscript). In addition, to facilitate the readers' understanding, we include the experimental schedule as Fig. S6A.

9. All figure legends need to include statements on number of repeat experiments, number of mice per group

We now include this information in each figure legend and in lines 661–664 of the revised manuscript. Indeed, each experiment was performed independently at least twice, with 5 mice per group.

In vitro studies:

10. The claim that Blautia wexlerae metabolites affect adipocyte differentiation is based on exposing the 3T3L1 pre-adipocyte cell line to supernatant from Blautia wexlerae cultures. I do think that this approach is too limited. In vivo, there is no direct contact between Blautia wexlerae, a gut commensal, and adipocytes. Any change in metabolites that is introduced by Blautia wexlerae in vivo is much more complex, taking into account changes in the gut microbiome, its metabolites, adipokines, gastrointestinal hormones and so on.

We agree with your concern regarding how *Blautia* affects physically distant organs. To avoid overinterpreting our results, we have reworded our conclusion (lines 258–260) and have addressed this limitation (lines 342–345 of the revised manuscript).

To understand the complex effects of administration of *B. wexlerae*, we performed additional experiments including measuring the serum levels of the GLP-1 gastrointestinal hormone (Fig. S14) and *Blautia*-derived metabolites (Fig. S23) and metabolic changes in muscle and liver (Fig. S13). In the Results, we show that GLP-1 and the metabolites S-adenosylmethionine, acetylcholine, and L-ornithine were either undetectable or were comparable among CD-fed mice, untreated HFD-fed mice, and *B. wexlerae*-treated HFD-fed mice (lines 271–274 and 327–339). Interestingly, adipose tissue and liver showed similar metabolic changes, in particular succinate accumulation, which is a metabolic signature of thermogenesis (lines 261–270). In addition, administration of *B. wexlerae* altered the intestinal environment including the composition of the gut microbiome and SCFA concentrations (Fig. 5, 6). Overall, it is important to understand that *B. wexlerae* likely offers various benefits for health maintenance through introducing these complex changes, as we now describe in lines 453–461 of the revised manuscript. Thank you again for your important insight.

11. The LC-MS/MS experiments were performed on epididymal MAF of mice suspended in phosphate-buffered saline or bacterial cultured medium. It is also not clear to me that phosphate-buffered is the correct control here – is there any reason not to use culture medium without bacteria or with other bacteria? However, my general concern is that in vivo there is presumably no direct contact between Blautia wexlerae, a gut commensal, and fat tissue.

We apologize for the unintentionally misleading sentences in the Methods section. Epididymal MAF was prepared as a suspension in PBS, and this suspension was processed for LC-MS/MS analysis. Samples of bacteria-cultured medium underwent the same processing protocol for LC-MS/MS. We have corrected this information in lines 815–823 of the revised manuscript. As mentioned earlier, we

understand that there is no direct contact between *B. wexlerae* and fat tissue and therefore interpret our results accordingly.

12. The claim that Blautia wexlerae metabolites confer anti-inflammatory effects to adipose tissue should be confirmed by ex vivo analysis of cytokine expression and analysis of immune cell composition and function. This is important as the authors specify reduced s100a8 mRNA expression (a chemokine recruiting macrophages) and reduced mRNA levels of TNFa in adipose tissue.

Accordingly, we performed the additional experiments. The administration of *B. wexlerae* decreased the number of macrophages, especially pro-inflammatory M1-like macrophages, that infiltrated into the eAT of HFD-fed mice (Fig. 2H, S10A, S10B); we describe these findings in lines 202–205 of the revised manuscript. Thank you again for your helpful comments.

REVIEWERS' COMMENTS

Reviewer #1 (Remarks to the Author):

The original review comments were diverse and the authors have done a thorough job in both responding to these comments and revising their manuscript. This remains a very interesting study with high quality work that is now presented in an even better revised written manuscript.

Reviewer #2 (Remarks to the Author):

There were many review points in this article and the authors made a reasonable effort to address most of them in the rebuttal and revised manuscript. They also perform additional experiments, in particular to determine the potential mechanisms that may underlie the physiological effects in tissues other than adipose tissue such as liver and muscle. The introduction and methodology sections have been significantly improved compared to the initial version.

However, I have still some issues with the representation of animal data. The explanations provided by the authors are not really reassuring. Some panels in the same figure included mice from 2 or 3 independent experiments without clear justifications. The fact that some data had to be removed due to blood hemolysis is mentioned but that should not have affected the weight of tissues. For example, why the weight of the animals in 2A would be the results of 3 independent experiments and the weight of the eWATs in 2C; the result of 2 experiments? Overall, I would have like to see better justification for the number of mice used in these experiments, especially considering the low number of animals used in general in this study.

Reviewer #3 (Remarks to the Author):

The authors have addressed all questions. One remaining issue is that the experiments were solely performed with male mice. Please examine the key protective effects of *Blautia wexleri* also with female mice.

National Institutes of Biomedical
Innovation, Health and Nutrition

**Laboratory of Vaccine Materials, Center for Vaccine and Adjuvant Research,
Laboratory of Gut Environmental System, Collaborative Research Center for
Health and Medicine**

**7-6-8 Asagi Saito, Ibaraki-shi, Osaka 567-0085, Japan
Tel: 81-72-641-9871, Fax: 81-72-641-9872**

7 July 2022

Dear Reviewers,

Thank you for reviewing our revised manuscript, entitled “Unique metabolic profiles of *Blautia wexlerae* achieve beneficial effects for the control of obesity and type 2 diabetes” (NCOMMS-21-43496). We appreciate the positive comments from the reviewers and have revised the manuscript accordingly again. Our point-by-point responses to your comments follow.

I respectfully request your reconsideration regarding the publication of our final revised manuscript in *Nature Communications*.

Sincerely yours,

Jun Kunisawa, Ph.D.
Director

Point-by-point response to the reviewers' comments

Reviewer #1 (Remarks to the Author):

The original review comments were diverse and the authors have done a thorough job in both responding to these comments and revising their manuscript. This remains a very interesting study with high quality work that is now presented in an even better revised written manuscript.

We appreciate your decision to accept the publication of our revised manuscript in *Nature Communications* and thank you very much again for your efforts in reviewing our manuscript.

Reviewer #2 (Remarks to the Author):

There were many review points in this article and the authors made a reasonable effort to address most of them in the rebuttal and revised manuscript. They also perform additional experiments, in particular to determine the potential mechanisms that may underlie the physiological effects in tissues other than adipose tissue such as liver and muscle. The introduction and methodology sections have been significantly improved compared to the initial version.

However, I have still some issues with the representation of animal data. The explanations provided by the authors are not really reassuring. Some panels in the same figure included mice from 2 or 3 independent experiments without clear justifications. The fact that some data had to be removed due to blood hemolysis is mentioned but that should not have affected the weight of tissues. For example, why the weight of the animals in 2A would be the results of 3 independent experiments and the weight of the eWATs in 2C; the result of 2 experiments? Overall, I would have like to see better justification for the number of mice used in these experiments, especially considering the low number of animals used in general in this study.

Thank you very much again for your efforts in reviewing our manuscript. In accordance with the reviewer's comments and instructions of the editorial office, mouse numbers/data has been clarified in the Figure legends, Methods and Source Data file. Because the minimum number of experiments is conducted in consideration of animal welfare, the number of experiments differs among panels. For example, we performed total 3 independent experiments related to Figure 2. The body weight of mice has been checked in all experiments, the weight of the eWATs was examined at first and second

experiments, and the third one was performed for the other experiments. In all experiments, we have performed at least 2 independent experiments for one panel and obtained the reproducible results.

Reviewer #3 (Remarks to the Author):

*The authors have addressed all questions. One remaining issue is that the experiments were solely performed with male mice. Please examine the key protective effects of *Blautia wexlerae* also with female mice.*

In diet-induced obesity and diabetic models, it is generally known that male mice are more susceptible to diet-induced weight gain and diabetic pathology (I. Casimiro, et al., Journal of Diabetes and its Complications, 35(2) 107795, 2021), and we accordingly used male mice in this study. Of course, we agree with your consideration about sexual differences and thus mentioned this point in Discussion section (lines 561-564) of the revised manuscript. We would like to examine the sexual differences of the effects of *B. wexlerae* in the future study including human.

Thank you very much again for your efforts in reviewing our manuscript.